# Mastering morphology of non-fullerene acceptors towards long-term stable organic solar cells

Kang An [1,6], Wenkai Zhong [1,2,6], Feng Peng [3,6], Wanyuan Deng[1], Ying Shang [1,4], Huilei Quan [1], Hong Qiu[1], Cheng Wang [5], Feng Liu [2], Hongbin Wu [1], Ning Li [1,4] ✉, Fei Huang [1,4] ✉ & Lei Ying [1] ✉

Despite the rapid progress of organic solar cells based on non-fullerene acceptors, simultaneously achieving high power conversion efficiency and long-term stability for commercialization requires sustainable research effort. Here, we demonstrate stable devices by integrating a wide bandgap electron-donating polymer (namely PTzBI-dF) and two acceptors (namely L8BO and Y6) that feature similar structures yet different thermal and morphological properties. The organic solar cell based on PTzBI-dF:L8BO:Y6 could achieve a promising efficiency of 18.26% in the conventional device structure. In the inverted structure, excellent long-term thermal stability over 1400 h under 85 °C continuous heating is obtained. The improved performance can be ascribed to suppressed charge recombination along with appropriate charge transport. We find that the morphological features in terms of crystalline coherence length of fresh and aged films can be gradually regulated by the weight ratio of L8BO:Y6. Additionally, the occurrence of melting point decrease and reduced enthalpy in PTzBI-dF:L8BO:Y6 films could prohibit the amorphous phase to cluster, and consequently overcome the energetic traps accumulation aroused by thermal stress, which is a critical issue in high efficiency non-fullerene acceptors-based devices. This work provides insight into understanding non-fullerene acceptors-based organic solar cells for improved efficiency and stability.

Organic solar cells (OSCs) have attracted widespread attention and considered as one of the most promising clean and renewable energy candidates due to their unique merits of light-weight, flexibility, and cost-efficient manufacturing with roll-to-roll processing techniques[1,2]. Recently, with the emergence of non-fullerene acceptors (NFAs), the power conversion efficiencies (PCEs) of OSCs have exceeded 18%, primarily due to their remarkable optical and electrical properties and favorable morphological compatibility with conjugated polymer donors[3–6]. Apart from PCEs, long-term stability also plays a critical role in determining the success of photovoltaic technology[7–9]. Generally, the stability of OSCs can be affected by various external surrounding stresses, including light, temperature, oxygen, water, mechanical stress, and so forth[10–13]. These stresses can lead to intrinsic instability of devices, especially the morphology changes of the active layer and the

[1]Institute of Polymer Optoelectronic Materials and Devices, State Key Laboratory of Luminescent Materials and Devices, South China University of Technology, Guangzhou 510640, China. [2]Frontiers Science Center for Transformative Molecules, Center of Hydrogen Science, and School of Chemistry and Chemical Engineering, Shanghai Jiao Tong University, Shanghai 200240, China. [3]South China Institute of Collaborative Innovation, Dongguan 523808, China. [4]Pazhou Lab, Guangzhou 510320, China. [5]Advanced Light Source Lawrence Berkeley National Laboratory, Berkeley, CA 94720, USA. [6]These authors contributed equally: Kang An, Wenkai Zhong and Feng Peng. ✉e-mail: ningli2022@scut.edu.cn; msfhuang@scut.edu.cn; msleiying@scut.edu.cn

diffusion of interfaces into the photoactive layer[14–16]. As the operating temperature of OSCs elevated upon continuous illumination, such thermal stress can detrimentally accelerate the evolution of active layer morphology. Hence, the metastable active layer morphology keeps a hindrance to simultaneously obtain the goal of high PCEs and long-term stability. We summarized the stability of previously published OSCs based on the state-of-the-art NFAs, Y6 and its derivatives, evaluated under various conditions (Supplementary Fig. 1, Supplementary Table 1). Nearly all devices encountered faster degradation during an operation span from 200 to 2400 h, even though few systems could be relatively stable under the dark condition without external stress.

Normally, due to the long-range disorder and short-range order properties of organic semiconductors as well as the complex mixed phase regions at the interface between polymer donor and NFAs, finely regulating the film morphology of active layer is still on their path to improve exciton dissociation and charge transportation[17,18]. Unfortunately, the bulk-heterojunction (BHJ) morphology of OSCs is kinetically frozen during solution processing, which is usually far from its thermodynamic equilibrium. Therefore, it gradually evolves to the equilibrium state, causing the suboptimal morphology[19,20]. Some valid strategies like using solid additives[21,22], designing oligomer molecular and polymer acceptors[23,24], establishing structure device[25], as well as introducing a third donor or acceptor component have been extensively explored to significantly suppress the burn-in losses for NFA-based OSCs[26]. However, almost no state-of-the-art OSCs could deliver a satisfied $T_{80}$ lifetime under thermal or illumination stress so far.

Therefore, material-related thermal properties were introduced to estimate morphological optimization and long-term stability[27,28]. By utilizing thermal factors, various NFAs can drive the morphology transitions from non-equilibrium towards equilibrium state via different approaches. If two or more similar NFAs with matched miscibility yet different thermal propriety and morphology are mixed together, a series of pseudo-new NFAs could be produced by varying the mixing ratios. Thus, it is of great importance to analyze their thermal behaviors and morphological changes as a function of the properties of NFAs, and to further explore the relationship between these factors and the long-term stability of efficient OSCs.

It has been reported that the morphological properties of organic materials can be obtained by using Ultraviolet–visible-near infrared (UV-vis-NIR) absorption and photoluminescence (PL) spectra to track the volume and nature of amorphous phases[29–31]. Based on the optical modelling, subtle morphological information could be attained, such as optimal exciton splitting and phase composition of the photoactive layer. In combination with the well-known X-ray scattering measurement identifying the ordered phases, a complete picture on the morphology of organic materials could be obtained. To construct efficient and stable OSCs, subtle morphology changes in BHJ should be carefully investigated since they are highly sensitive to the composition of active layer and various surrounding stresses. Within the framework of polymer:NFAs interactions, the amount of clustered amorphous phase can be monitored by corresponding optical signals. The relative intensity of PL value at NIR region is the illustration of amorphous clusters behavior of NFAs and can be used as an indicator to backtrack the subtle morphology changes in active layer. It is advised to construct efficient and long-term stable OSCs via integrating two similar NFAs together to systematically regulate the thermal behavior and morphology.

Here, we adopt the aforementioned strategy by employing two state-of-the-art NFAs (L8BO and Y6) featuring similar chemical structure (shown in Fig. 1a), yet different thermal properties and morphologies, with a polymer donor PTzBI-dF containing an alternating benzo[1,2-*b*:4,5-*b'*]dithiophene (BDT) and pyrrolo[3,4-*f*]benzotriazole-5,7(6*H*)-dione (TzBI) structure unit[32–34]. The PTzBI-dF:L8BO:Y6 devices exhibit a promising PCE of 18.26% and excellent stability.

Nearly 95% of original efficiency could be maintained under continuous heating at 85 °C for 1400 h. Systematic investigation with grazing incidence wide-angle X-ray scattering (GIWAXS) and optical spectra revealed that tuning the NFA combinations of L8BO and Y6 not only gradually regulated film morphology, but also prevented the amorphous phase aggregation in BHJ thin film. The robust film morphology suppressed traps accumulation and charge recombination in PTzBI-dF:L8BO:Y6 based devices, leading to optimized OSCs with substantially improved device efficiency and thermal stability.

## Results

### Materials properties and photovoltaic performance

The chemical structures of PTzBI-dF, L8BO and Y6 are shown in Fig. 1a. The chemical difference between L8BO and Y6 is the branched alky side chains, which results in similar but slightly distorted intermolecular arrangement. GIWAXS gives the morphology information of L8BO:Y6 blend films with different ratios. The quality of crystallization can be evaluated by the crystalline coherence length (CCL), which is calculated through fitting the averaged *I-q* curves obtained from GIWAXS and using the Scherrer equation (Fig. 1b and Supplementary Fig. 2). A higher CCL represents a lower degree of imperfections and dislocations that can interfere the molecular packing in crystallites[35]. All films show lamellar stacking reflections at low *q* region ($q$ -0.29 Å$^{-1}$ for PTzBI-dF, $q$ -0.42 Å$^{-1}$ for L8BO, and $q$ -0.30 Å$^{-1}$ for Y6) in the in-plane direction and pronounced π−π stacking reflection at high *q* position of -1.74 Å$^{-1}$ in the out-of-plane direction, indicating all these materials adopt a preferential face-on orientation, which is beneficial to the charge transport. Notably, the π−π stacking reflection for Y6 is stronger and sharper than that of L8BO, indicating a higher degree of molecular order. The CCLs of L8BO and Y6 are 14.6 and 21.5 Å respectively, signifying the better crystalline quality of Y6. By mixing two NFAs with different ratios, the CCLs of L8BO:Y6 blend films located between those two neat films, suggesting that the morphology of NFAs could be finely manipulated.

To reveal the relevant optical properties of the L8BO:Y6 blends, UV-vis-NIR absorption spectra of L8BO:Y6 blends were measured and analyzed (Fig. 1c)[36,37]. The blend ratios have a strong influence on the relative strength of $A_O$, $A_A$ and $A_2$ peaks. The peak intensity contrast is shown in Fig. 1d. Here, the values of $A_O/A_A$ are flatten compared to the values of $A_O/A_2$ and $A_A/A_2$, which monotonically decreased with the blend ratio of L8BO and Y6. Such results reinforced that morphology can be fine-tuned by changing the blend ratio of NFAs, in agreement with the GIWAXS results. The maximum absorption peaks of PTzBI-dF, L8BO, and Y6 are located at 638, 797, and 830 nm, respectively, showing well complementary absorption in the range of 350−900 nm (Supplementary Fig. 4a). The absorption coefficients of blend films were also measured (Supplementary Fig. 4b). Compared to the PTzBI-dF:L8BO blend film, the introduction of Y6 expands the absorption range to the near-infra region, which helps to promote the photon harvesting. Differential scanning calorimeter (DSC) measurements were performed to elucidate the thermal properties of L8BO:Y6 mixtures. No detectable meting peaks could be observed in the second DSC profiles (Supplementary Fig. 5a). The glass transition temperature ($T_g$) was also difficult to directly obtained from DSC profiles. Therefore, the thermal behavior of the NFAs was evaluated using the melting point ($T_m$) and melting endotherm ($\Delta H$) extracted from the first heating scan. The DSC first heating curves for L8BO:Y6 blends with various ratios are depicted in Supplementary Fig. 5b. The $T_{m1}$ corresponding to L8BO gradually decreased from 320 °C to 290 °C. The $T_{m2}$ related to Y6 pinned at about 280 °C, suggesting the melting point could be easily affected even mixed with small amounts of L8BO (Supplementary Fig. 5c). The corresponding $\Delta H$ values of L8BO:Y6 can be determined by the integration of the melting peaks. The $\Delta H$ showed a sharp decline to 18.32 and 16.64 J g$^{-1}$ at moderate L8BO:Y6 ratios (0.7:0.5 and 0.5:0.7) compared to L8BO (43.83 J g$^{-1}$) and Y6 (28.59 J g$^{-1}$)

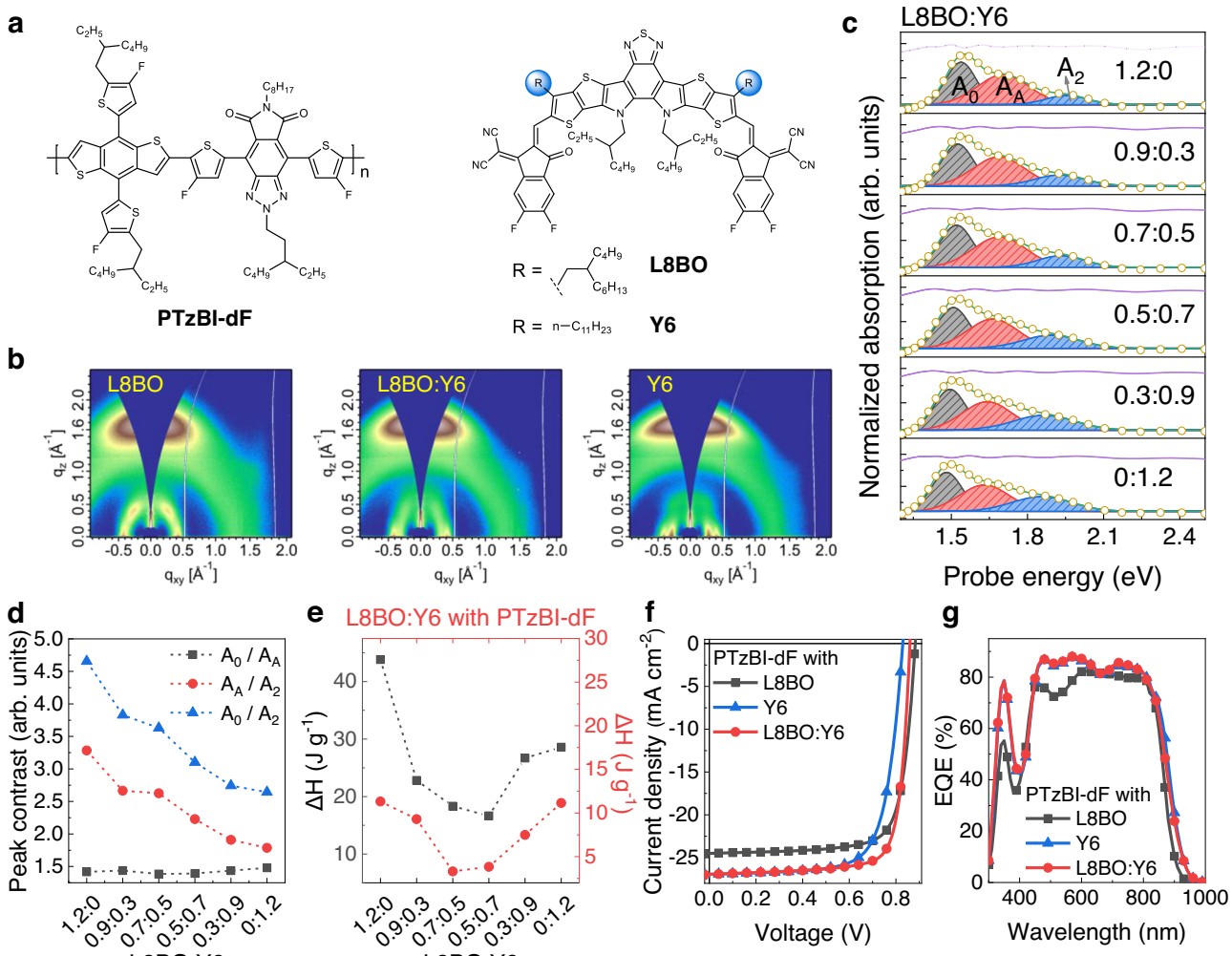

**Fig. 1 | Material properties and photovoltaic performance of optimized OSCs.**
**a** The chemical structures of polymer donor and non-fullerene acceptors (NFAs).
**b** GIWAXS images of the NFA thin films. **c** The decomposition details of ultraviolet–visible–near infrared (UV-vis-NIR) absorption spectra for L8BO:Y6 blends with various ratios; ordered phase indicate with $A_0$; amorphous phase indicate with $A_A$; higher electronic transitions indicate with $A_2$; The term 'arbitrary units' abbreviated as 'arb. units'. **d** The peak intensity contrast obtained from the UV-vis absorption spectra. **e** The melting endotherm ($\Delta H$) of obtained from differential scanning calorimetry (DSC); black dots indicate L8BO:Y6-related $\Delta H$; red dots indicate PTzBI-dF:L8BO:Y6-related $\Delta H$. **f** Current density–voltage (J–V) characteristics of devices under simulated AM 1.5 G irradiation (100 mW cm⁻²). **g** EQE spectra (PTzBI-dF:L8BO = 1:1.2, PTzBI-dF:Y6 = 1:1.2, PTzBI-dF:L8BO:Y6 = 1:0.7:0.5).

neat films (Fig. 1e). Hence, the eutectic point could be reached at these L8BO:Y6 ratios, where a $T_m$ depression and lower enthalpy are appeared at the meantime[38]. L8BO and Y6 showed different thermal diffusion into PTzBI-dF, determined with time-of-flight secondary ion mass spectrometry (TOF-SIMS) (Supplementary Fig. 8)[39]. To explore the thermal behavior of L8BO:Y6 when combining with polymer PTzBI-dF, DSC measurements for neat PTzBI-dF film and PTzBI-dF:L8BO:Y6 blend films were performed (Supplementary Fig. 5d, e). No detectable peaks could be observed in PTzBI-dF curves, but PTzBI-dF:L8BO:Y6 blends exhibited a broadened melting transition due to the NFAs. Therefore, it is reasonable to analyze the thermal properties of PTzBI-dF and the two NFAs through the heat flow $\Delta H$ characters. The $\Delta H_{mix}$ exhibits most suppress with 6.07 J g⁻¹ in 1:0.7:0.5 blend, indicating that the $\Delta H$ can be minimized through constructing two NFAs with different thermal properties (Supplementary Fig. 6). We focus on the blend ratio of 1:0.7:0.5 PTzBI-dF:L8BO:Y6 blends, as the mixed NFAs with similar structure may form a homogeneous amorphous mixture in blends, which also offers the best photovoltaic performance (cf. discussion on device performance below)[40–42].

The ionization potential was obtained with the photoelectron yield spectra, and HOMO energy levels of PTzBI-dF, L8BO, L8BO:Y6 and Y6 neat films were determined to be −5.51 eV, −5.93 eV, −5.95 and −5.99 eV, respectively (Supplementary Fig. 9). The LUMO energy level was estimated by subtracting the HOMO energy levels with the optical bandgaps of corresponding films. These results demonstrated that L8BO with a relatively shallow LUMO energy level has a higher potential to attain a large open-circuit voltage ($V_{OC}$). Collectively, integrating two NFAs with various ratios can effectively control the materials properties associated with thin-film morphology and molecular thermal behaviors.

To evaluate the photovoltaic performance and device stability, OSCs were constructed with a structure of indium-tin-oxide (ITO)/ PEDOT:PSS/active layer/PNDIT-F3N-Br/Ag[43]. The representative current density–voltage (J–V) curves and related parameters are summarized in Fig. 1f and Table 1, respectively. The PTzBI-dF:L8BO based device exhibited a PCE of 16.74%. The PTzBI-dF:Y6 device exhibited a high $J_{SC}$ of 26.94 mA·cm⁻² but a reduction in FF and $V_{OC}$, giving a PCE of 16.23%. The optimized PTzBI-dF:L8BO:Y6 device delivered the best PCE of 18.26% (Supplementary Fig. 10, Supplementary Table 5). Effective charge transport can be obtained in PTzBI-dF:L8BO:Y6-based devices since slightly enhanced external quantum efficiency (EQE) response as well as the similarly integrated $J_{SC}$ values to PTzBI-dF:Y6

**Table 1 | Photovoltaic parameters of PTzBI-dF:L8BO, PTzBI-dF:Y6 and PTzBI-dF:L8BO:Y6 based OSCs under simulated AM 1.5 G irradiation (100 mW cm$^{-2}$)**

| Blends | $V_{OC}$ (V) | $J_{SC}$ (mA cm$^{-2}$) | $J_{SC, EQE}$ (mA cm$^{-2}$) | FF (%) | PCE (%) |
|---|---|---|---|---|---|
| PTzBI-dF:L8BO | 0.882 (0.882 ± 0.001) | 24.51 (24.20 ± 0.31) | 24.12 | 76.49 (75.76 ± 0.73) | 16.74 (16.49 ± 0.25) |
| PTzBI-dF:Y6 | 0.829 (0.829 ± 0.001) | 26.94 (26.45 ± 0.49) | 26.45 | 72.66 (72.12 ± 0.54) | 16.23 (16.03 ± 0.20) |
| PTzBI-dF:L8BO:Y6 | 0.860 (0.860 ± 0.001) | 26.95 (26.56 ± 0.39) | 26.37 | 78.78 (77.65 ± 1.13) | 18.26 (17.95 ± 0.31) |

based devices in the wavelength range of 450–850 nm (Fig. 1g). Hole ($\mu_h$) and electron ($\mu_e$) mobilities were examined through the space-charge-limited current (SCLC) measurement (Supplementary Fig. 11)[44]. Enhanced and balanced charge mobilities were exhibited in PTzBI-dF:L8BO:Y6 OSCs, suggesting that the hole and electron transport channels can be finely optimized by regulating the morphology of NFAs. Dark $J$-$V$ characteristics exhibit the slightly lower leakage current in the low voltage regime, corresponding to a higher shunt resistance for PTzBI-dF:L8BO:Y6 device (Supplementary Fig. 10b)[45]. The charge recombination was investigated using the relationship of $V_{OC}$ and $J_{SC}$ as a function of light intensity ($P_{light}$) (Supplementary Fig. 10c, d). Effectively enhanced charge carriers transportation and inhibited recombination were exhibited in PTzBI-dF:L8BO:Y6-based devices[46,47]. Therefore, fine-tunning the thermal and morphological properties in PTzBI-dF:L8BO:Y6 was critical to modify charge transport as well as enhance $J_{SC}$ and FF in devices.

Fourier-transform photocurrent spectroscopy (FTPS) and electroluminescence external quantum efficiencies ($EQE_{EL}$) were measured to identify the $E_{loss}$ (Supplementary Fig. 13, Supplementary Table 8)[48]. The difference of $E_{loss}$ in above three devices mainly stems from the non-radiative recombination energy loss ($\triangle E_3$) values and the total $E_{loss}$ of PTzBI-dF:L8BO:Y6-based devices was determined by the lowest energy loss in two NFAs. The Urbach energy ($E_U$) can be fitted from the low photon energy range of EQE spectra. The PTzBI-dF:L8BO:Y6 blend showed low energetic disorder, enabling efficient charge generation. In this respect, controlling the thermal and morphological properties of NFAs in BHJ provides an effective strategy to alleviate energy loss and improve $J_{SC}$ and FF values. All blend films present uniform and well distributed film morphologies, indicating good miscibility of components, as shown in the transmission electron microscopy (TEM) images and atomic force microscopy (AFM) (Supplementary Fig. 14 and 15).

**Thermal stability of OSCs and film morphology**

In order to investigate the effect of NFAs morphology on the long-term stability of OSCs, thermal stability measurement was performed in a nitrogen-filled glovebox and all devices heated continuously on a hot plate. All conventional devices encountered similarly undesirable degradation to less than 80% within 200 h, indicating the intrinsic instability caused by PEDOT:PSS buffer layers (Supplementary Fig. 16)[49]. The inverted structure has been proven to prevent the potential degeneration factors from interfaces[50,51]. The photovoltaic performance of PTzBI-dF:L8BO, PTzBI-dF:Y6 and PTzBI-dF:L8BO:Y6-based inverted devices lags behind those of the conventional structure counterparts (Supplementary Fig. 17, Supplementary Table 9). However, slightly higher efficiencies can still be achieved in PTzBI-dF:L8BO:Y6-based devices. Initially, the thermal stability was evaluated with a temperature of 65 °C (Supplementary Fig. 18). The normalized photovoltaic parameters are plotted with respect to thermal ageing time. Both PTzBI-dF:L8BO and PTzBI-dF:Y6-based devices maintained good stabilities, with the PCEs over 90% after the thermal stress over 500 h. Note that there was almost no significant fluctuation for PTzBI-dF:L8BO:Y6 devices after thermally annealed, indicating robust stability for the blends. To further study the superior thermal stability of PTzBI-dF:L8BO:Y6, annealing temperature of 85 °C was used. As shown

in Fig. 2, thermal ageing at 85 °C led to fast degenerated photovoltaic parameters than those at 65 °C for both PTzBI-dF:L8BO and PTzBI-dF:Y6 devices. The FF exhibited the biggest drop, while the $V_{OC}$ and $J_{SC}$ remained relatively robust, indicating degradation predominantly occurred in the mixed region. Compared to the severe PCE decay of ~19% and ~23% for PTzBI-dF:L8BO and PTzBI-dF:Y6-based devices, the PTzBI-dF:L8BO:Y6 device retained over 95% of its original PCE after being thermally annealed for about 1400 h, indicating excellent thermal properties of the blends. Acceptor alloys could be formed in L8BO:Y6, because the similar structure of L8BO and Y6, and a linear dependence of $V_{OC}$ varied with the stoichiometry of L8BO and Y6 in OSCs (Supplementary Table 5). The cyclic voltammetry (CV) measurement was used to further determine the energy levels of L8BO:Y6. When L8BO and Y6 blended, the mixed acceptors show shifted energy levels dependent on the components (Supplementary Fig. 19, Supplementary Table 10), which reinforced the formation of effective alloy for L8BO:Y6 acceptors[52]. Commercially available polymer donors of D18 and PM6 were utilized to further investigate the stability improvement with the robust combination of L8BO:Y6 acceptors. The L8BO:Y6 combination led to improved thermal stability when combined with D18 and PM6 polymer donors than those for reference devices (Supplementary Fig. 21, 22). The degeneration of FF was effectively inhibited. The D18:L8BO:Y6 and PM6:L8BO:Y6 devices showed the decay of ~10% and ~20% of its original PCE after being thermally annealed for about 500 h, indicating a good improvement in thermal stability. Hence, it was a feasible strategy to utilize acceptor alloys strategy to improve the stability of OSCs.

To gain deep insight into the control of morphology by regulating NFAs, GIWAXS was performed on PTzBI-dF:L8BO:Y6 blends before and after thermal aging at 85 °C for 100 h. The relevant averaged curves of the out-of-plane (OOP) and in-plane (IP) directions are shown in Figs. 3a and 3b, respectively. For fresh batches, all blends showed similar characteristics with π-π stacking peaks at ~1.77 Å$^{-1}$ in the OOP direction and lamellar peaks at ~0.3 Å$^{-1}$ in the IP direction. These are combined signals from both polymer donor and NFAs. The PTzBI-dF:L8BO showed a relatively broad π–π reflection from 1.5 to 2.0 Å$^{-1}$ (Supplementary Fig. 23). The addition of Y6 led to narrowed and intensified π–π stacking reflections in PTzBI-dF:L8BO:Y6 films, indicating better molecular packing[53]. Each blend was estimated by CCLs of the lamellar stacking reflections (Supplementary Table 12). The CCL was calculated to be 49.36 Å for the PTzBI-dF:L8BO blends and gradually increased with the load of Y6 contents, up to 69.60 Å for PTzBI-dF:Y6 counterparts. Meanwhile, resonant soft X-ray scattering (RSoXS) was carried out at a beam energy of 285.0 eV to disclose the morphology of fresh blends (Supplementary Fig. 25). All blends showed similar scattering profiles with broad humps within the probed $q$ range, giving correlation lengths from 10–13 nm in PTzBI-dF:L8BO:Y6 blends, as fitted by the correlation length model[44]. Hence, by blending L8BO and Y6, the CCLs in BHJ thin films can be regulated while maintaining phase-separated length scales, offering an effective method to finely adjust the morphology and photophysical properties of these blends. For aged GIWAXS blends, it is clearly shown that PTzBI-dF:L8BO:Y6 blends exhibit no significant changes in both IP and OOP directions, which agrees well with the long-term thermal stability of PTzBI-dF:L8BO:Y6. Nevertheless, no clear changes could be

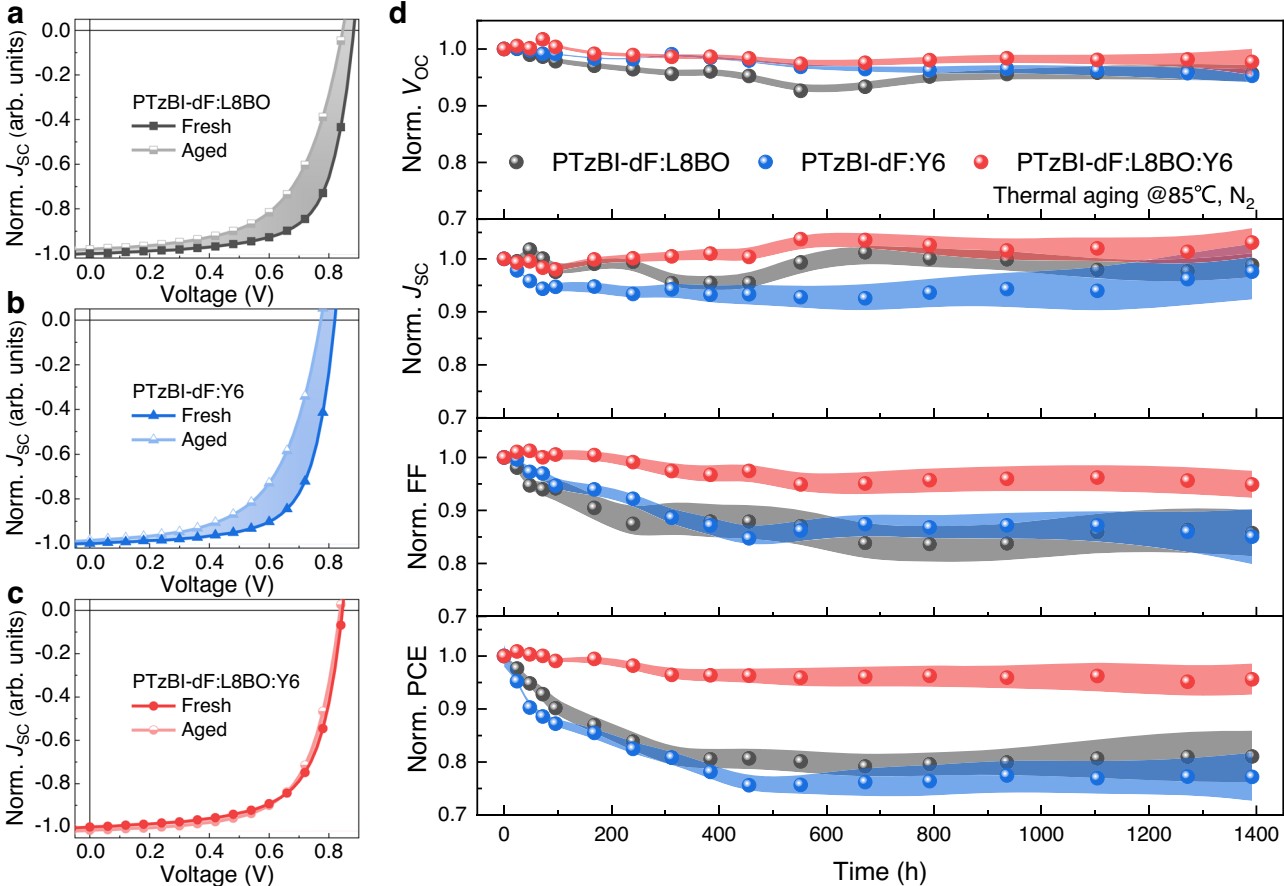

**Fig. 2 | The device thermal stability under continuous 85 °C ageing in a dry nitrogen atmosphere without encapsulation. a–c** Recorded *J–V* curves normalized with initial short-circuit current density; shaded areas indicate the gradually degeneration of *J–V* curves. **d** Normalized photovoltaics parameters as a function of time over four devices to PTzBI-dF:L8BO, PTzBI-dF:Y6 and PTzBI-dF:L8BO:Y6 photoactive layers; shaded areas indicate the error bar of photovoltaics parameters.

observed for PTzBI-dF:L8BO and PTzBI-dF:Y6 based blend films, indicating that the main loss of FF may not arouse by the disturbance of order phase in BHJ. To precisely determine the effect of thermal stress related to the order phase, CCLs of lamellar and π-π stacking were quantified and summarized in Table 12 (Supplementary). The CCLs in all aged thin films encounter few turbulences (<5%) compared to the fresh batch, conflicting to the FF burn-in loss within 100 h. Hence, the degradation of PTzBI-dF:L8BO and PTzBI-dF:Y6 based OSCs should not be directly related to the ordered phase[49].

Optical spectroscopy was also used to determine the relevant structural properties of organic semiconductor films. With the model of weak excitonic coupling, it has proven that band positions, shapes and relative intensities are relevant to ordered phase, amorphous phase and amorphous cluster in films[29–31,36,37]. Hence, the morphological decay under thermal stress can also be analyzed with the optical profiles. Due to spectral overlap between amorphous phase and ordered phase, there is efficient resonance energy transfer (RET) from amorphous towards ordered regions. However, if larger clusters of the amorphous phase are present which exceed the typical length scale of RET, then a significant amount of PL from the amorphous phase can be observed. Assuming similar PL quantum yields ($\varnothing_{PL}$) in the ordered and amorphous phases, the relative energy transfer quantum yield from the amorphous to the ordered phase ($\varnothing_{RET} / \varnothing_{PL}$) can be used as a probe to detect the occurrence of larger clusters of the amorphous phase in films with the sketch shown in Supplementary Fig. 26. Generally, the morphology can be divided into donor/acceptor-dominated phases and inter-mixed phase. Here, we cannot distinguish the

accurate aggregates' location according to the common phase division, considering the interpenetrated network of BHJ and highly quenched PL intensity of PTzBI-dF in blends. Nevertheless, the optical analysis can still represent the sum of aggregates' information from the amorphous phase. To evaluate the amorphous phase before and after thermal stress, we measured the UV-vis-NIR and PL spectra of L8BO, Y6 and L8BO:Y6 acceptor films and its corresponding BHJ combined with PTzBI-dF polymer donor before and after thermal ageing. (Fig. 3c and Supplementary Fig. 27–29). Here, the $\varnothing_{RET} / \varnothing_{PL}$ of pristine acceptor films only showed slightly fluctuation before and after thermal ageing (Supplementary Table 17). Note that both PTzBI-dF:L8BO and PTzBI-dF:Y6 blends exhibited unexcepted drop of $\varnothing_{RET} / \varnothing_{PL}$, which decreased from 0.39 to 0.19 and from 0.34 to 0.04, respectively, indicating large amorphous cluster in films may occurred in two blends. Considering the obvious overlap of L8BO and Y6 optical spectrum, potentially RET between two NFAs may disturb the information obtained from the PTzBI-dF:L8BO:Y6 film. TEM and RSoXS characterizations were performed to evaluate the domain change in aged blends. Both PTzBI-dF:L8BO and PTzBI-dF:Y6 blends exhibited enlarged distribution of bright and dark regimes as well as obviously changed humps within the probed *q* range (Supplementary Fig. 30), which supported large amorphous cluster in films observed in optical spectrum, combined with the nearly identical GIWAXS patterns. The PTzBI-dF:L8BO:Y6 blend exhibited stable film morphology. Hence, the amorphous clustering in PTzBI-dF:L8BO and PTzBI-dF:Y6 blends upon long-term annealing could induce the decay of device FF. Combined with the greatly suppressed ΔH in PTzBI-dF:L8BO:Y6 blends, the mixed

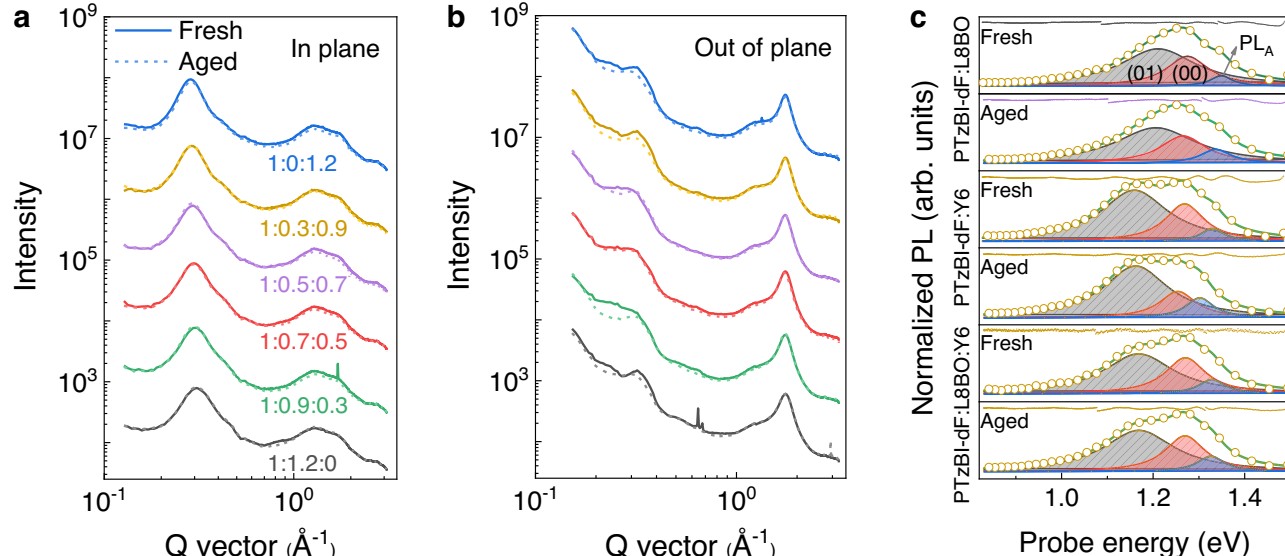

**Fig. 3 | Analysis of BHJ morphology of fresh and aged blend films (under 85 °C for 100 h in N₂ atmosphere).** GIWAXS line cuts of fresh and aged blend films based on different acceptors. **a** in-plane (IP) direction and (**b**) out-of-plane (OOP) direction. **c** PL profiles of fresh and aged PTzBI-dF:L8BO, PTzBI-dF:Y6 and PTzBI-dF:L8BO:Y6 blends (from top to bottom); the spectral decomposition details shown in methods.

NFAs in PTzBI-dF:L8BO:Y6 blends showed the restrained aggregation, which was favorable for achieving stable morphology on both order and amorphous regions, and thus, improved device stability.

## Charge carrier dynamics of fresh and aged OSCs

To evaluate the effect of regulated film morphology on charge carrier dynamics, a series of characterizations were performed to understand the degradation in FF under thermal stress. Impedance spectroscopy measurement (Supplementary Fig. 32, Supplementary Table 18) show that the PTzBI-dF:L8BO:Y6-based device encountered a slightly variation of transport ($R_0$) and recombination resistance ($R_1$) before and after thermal ageing, subjecting to PTzBI-dF:L8BO and PTzBI-dF:Y6 counterparts[54]. The $J_{ph} - V_{eff}$ curves (Fig. 4a–c) were used to extract stable P (E, T) values for PTzBI-dF:L8BO:Y6-based device, in contrast to those with severe decrease for PTzBI-dF:L8BO and PTzBI-dF:Y6 blends. The transient photocurrent (TPC) and transient photovoltage (TPV) measurements (Supplementary Fig. 33–34) indicate that the charge extraction time ($\tau_{TPC}$) and photocarrier lifetime ($\tau_{TPV}$) did not change in PTzBI-dF:L8BO:Y6-based OSCs[55]. Conversely, increase in $\tau_{TPC}$ and reduction in $\tau_{TPV}$ occurred simultaneously in PTzBI-dF:L8BO and PTzBI-dF:Y6-based OSCs. These combined results demonstrate that charge transport and collection were barely influenced on optimized PTzBI-dF:L8BO:Y6 film via fine-tunning thermal and morphology properties after long-term thermal stresses. Energetic disorder is also a common issue in OSCs due to the semicrystalline properties of organic semiconductor materials. The capacitance–voltage/frequency ($C - V/\omega$) measurements were used to examine the trap density difference aroused from the regulation of thermal and morphology properties in fresh and aged OSCs (Fig. 4d–i, Supplementary Fig. 35–36)[56,57]. The trap density in aged OSCs based on PTzBI-dF:L8BO:Y6 maintained nearly identical to the fresh samples. However, substantial trap accumulation was observed in PTzBI-dF:L8BO and PTzBI-dF:Y6 devices, suggesting that amounts of excitons and carriers were trapped in the tail states. The DOS center position of PTzBI-dF:L8BO:Y6 moving towards higher energy may related to the changed interface properties in BHJ[58]. Collectively, the undesirable degradation of FF in PTzBI-dF:L8BO and PTzBI-dF:Y6 devices could be ascribed to the aggregated amorphous cluster, and thus, increasing trap density and deteriorating charge transportation and recombination[50]. Controlling the thermal and morphological properties by two NFAs

successfully demonstrated robust morphological stability and prohibited the accumulation of trap density, enabling much stable FFs and PCEs when combining the NFAs with the polymer PTzBI-dF.

## Discussion

This work demonstrates the systematic regulation of the thermal behavior and morphology properties of NFAs, and reports an efficient strategy to simultaneously improve the PCE and thermal stability of NFA-based OSCs. The melting endotherm (ΔH) can be minimized through constructing two NFAs possessing different thermal properties with polymer donors. Meanwhile, the film morphology can be gradually tuned as a function of the NFA properties. The combined regulation of thermal and morphological properties enables us to explore the relationship between these factors and the long-term stability of efficient OSCs.

In view of evaluating active layers before and after continuous thermal stress, the morphological features were extracted by GIWAXS and optical modelling. A complete picture on the morphology of organic materials could be obtained, as illustrated in Fig. 5. The order phase in all blends encountered few turbulences as obtained from the GIWAXS results. The amorphous phases in the PTzBI-dF:L8BO:Y6 film could be effectively prohibited into cluster under thermal stress, which occurred in PTzBI-dF:L8BO and PTzBI-dF:Y6 films probed by optical spectroscopy. The stable amorphous phases in the blend films are crucial to prevent trap accumulation, and as a result the fill factor degradation.

In summary, two NFAs (LB8O and Y6) with similar structure, yet different thermal properties and microscopic stacking morphologies were used to regulate the morphology of BHJ photoactive layer. By combining the optimized ratio of L8BO:Y6 with the polymer donor PTzBI-dF, BHJ OSCs achieved a promising PCE of 18.26%. More importantly, the optimized OSCs could deliver excellent long-term thermal stability under 85 °C for 1400 h, which addresses the inherent thermal instability issues in state-of-the-art NFAs. The simultaneously obtained high performance and thermal stability originated from the balanced CCL and retained robust amorphous phase in the BHJ films. Electrical characterizations and analyses revealed that the optimized BHJ morphology could also overcome the unfavorable charge transport and charge recombination induced by amorphous clustering, leading to superior device

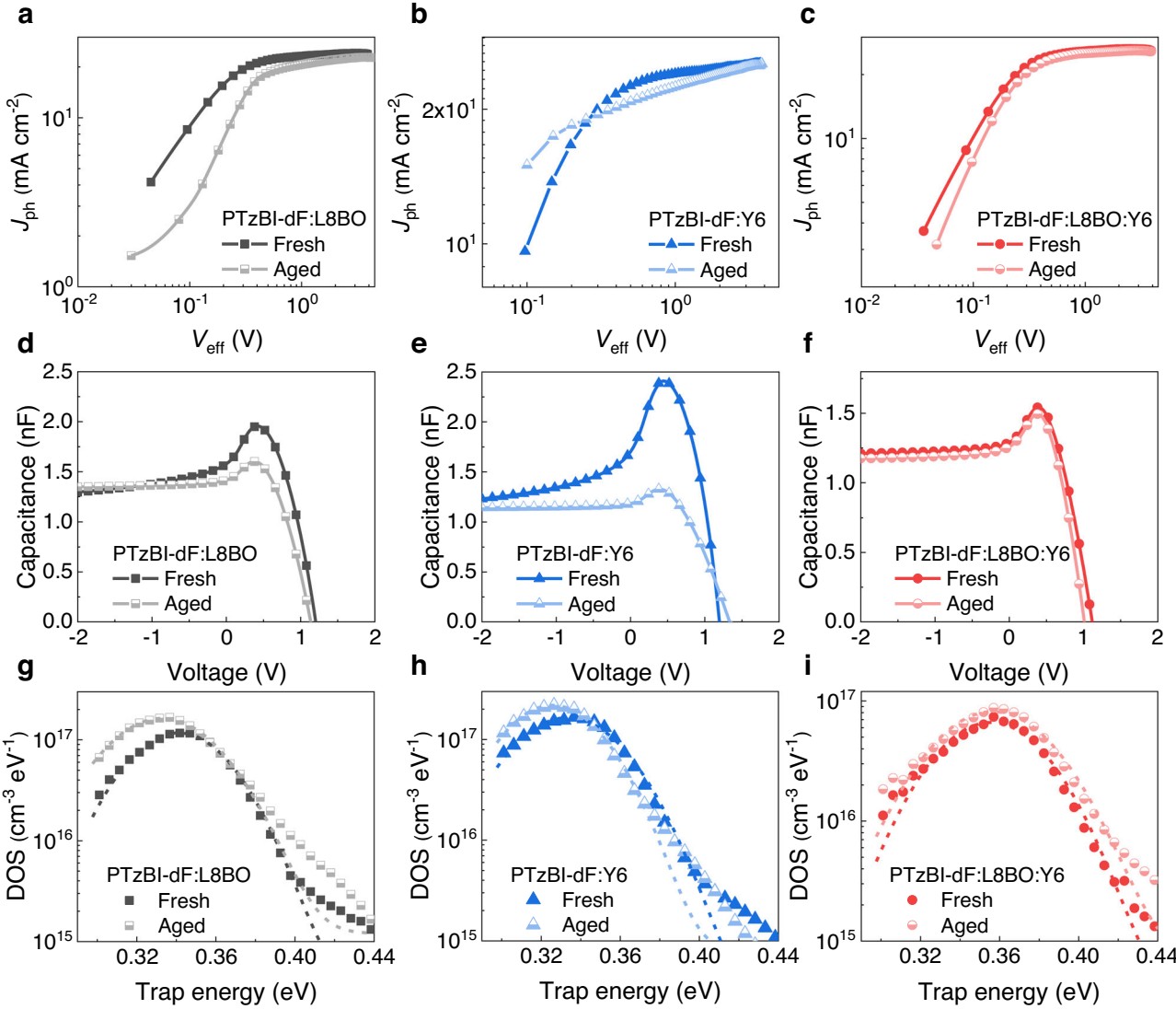

**Fig. 4 | Charge carrier dynamics of fresh and aged OSCs after 85 °C ageing. a–c** $J_{ph}$-$V_{eff}$ characteristics, **d–f** capacitance versus applied voltage and, **g–i** defects density of state for fresh and aged PTzBI-dF:L8BO, PTzBI-dF:Y6 and PTzBI-dF:L8BO:Y6 OSCs.

stability under thermal conditions. The findings demonstrated in this work provide insight into thin film morphological control to simultaneously optimize the performance and stability of organic photovoltaic devices.

## Methods

### Materials

PTzBI-dF, PM6, D18, L8BO, Y6 and PNDIT-F3N-Br were purchased from Volt-Amp Optoelectronics Tech. Co., Ltd, Dongguan, China. The aqueous dispersion of poly(3,4-ethylenedioxythiophene)-poly(-styrenesulfonate) (PEDOT:PSS, P VP AI 4083) was purchased from Heraeus, Germany. All solvents and chemicals were ordered from Sigma-Aldrich.

### Device fabrication

The conventional organic solar cells (OSCs) were manufactured with device structure of ITO/PEDOT:PSS/Active layer/PNDIT-F3N-Br/Ag. Patterned ITO substrates were ultrasonic cleaned with detergent, deionized water, and isopropanol for 30 min, respectively. Subsequently, cleaned ITO substrates kept in an electric thermostatic drying oven at 65 °C overnight. After treated with vacuum plasma cleaning of 120 s, PEDOT:PSS (about 35 nm), filtered with 13 mm/0.22 μm aqueous

polyethersulfone membrane syringe filters, was spin-coated on the precleaned ITO substrate with 3000 r/min for 30 s. Then, the PEDOT:PSS layer was baked at 150 °C for 15 min in air. The used active layer of PTzBI-dF:NFAs (1:1.2, wt:wt) were dissolved in chloroform (CF) with the total concentration of 12 mg mL$^{-1}$ and stirred at 50 °C for 5 h in a $N_2$ filled glove box. Then, the dibenzyl ether (DBE) additive was added into the solution with the volume ratio of 0.3%. The film thickness controlled at about 100–110 nm determined by a Bruker Dektak XT profilometer. The fabricated active layer was followed with thermal annealing at 100 °C for 5 min. About 5 nm of PNDIT-F3N-Br thin layer was coated (2000 rpm) onto the active layers with the concentrations of 0.5 mg mL$^{-1}$ dissolved with methanol solvents. Finally, silver electrodes (100 nm) were coated at the top of PNDIT-F3N-Br in a thermally depositing chamber with the vacuum degree of $2 \times 10^{-7}$ mbar. The effective area was 0.0516 cm$^2$ and further defined as 0.04 cm$^2$ with a non-refractive mask. OSCs with the inverted device structure of ITO/ZnO/Active layer/MoO$_x$/Ag was constructed to test the long-term stability of active layer. It avoided the effect of hygroscopicity, acidity, and anisotropic charge injection properties of PEDOT:PSS layers, which could result in device instability. A ZnO electron transport layer with a thickness about 30 nm was prepared on ITO substrates by spin-coating at 4000 rpm for 60 s from a ZnO precursor solution (zinc

PTzBI-dF:L8BO (Y6)

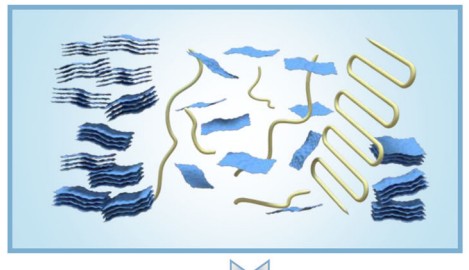

PTzBI-dF:L8BO:Y6

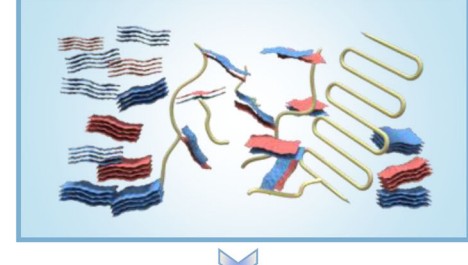

Aggregated amorphous cluster

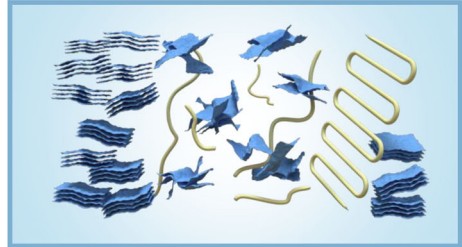

Restrained crystallization and aggregation

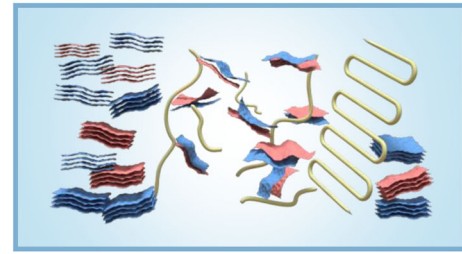

**Fig. 5 | Schematic representation of morphology comparation of the mixed region for PTzBI-dF:L8BO/PTzBI-dF:Y6 and PTzBI-dF:L8BO:Y6 blends under thermal stress.** In PTzBI-dF:L8BO and PTzBI-dF:Y6 blends, amorphous cluster regions might be formed as a result of NFAs aggregation, which act as morphological traps for electron transport and hole/electron recombination. It caused to long-term FF degradation. In contrast, PTzBI-dF:L8BO:Y6 blends exhibited robust amorphous phase benefiting from the restrained aggregation movements and modified BHJ morphology via employing L8BO and Y6 NFAs.

acetate dihydrate, 0.1 g mL⁻¹ solution in ethylene glycol monomethyl ether with 2.8 vol% of ethanolamine), followed by thermal annealing at 200 °C for 1 h. The PTzBI-dF-based active layers deposited as mentioned above. The used active layer of D18:NFAs and PM6:NFAs (1:1.2, wt:wt) were dissolved in chloroform (CF) with the total concentration of 11 and 15 mg mL⁻¹. The film thickness controlled at about 100–110 nm. The fabricated active layer was followed with thermal annealing at 100 °C for 5 min. At the vacuum degree of $2 \times 10^{-7}$ mbar, a thin layer (10 nm) of $MoO_x$ was thermally deposited as the anode interlayer, followed by thermal deposition of 100 nm Ag as the top electrode through a shadow mask.

**Charge-only devices fabrication**

The structure for hole-only and electron-only devices were ITO/PEDOT:PSS/Active layer/$MoO_x$/Ag and ITO/ZnO/Active layer/PNDIT-F3N-Br/Ag, respectively. The fabricating process of charge-only devices were similarly to the methods discussed above. Mobilities are illustrated with the Mott-Gurney equation $J = 9\varepsilon_0\varepsilon_r\mu V^2/8d^3$, where $J$ is the space charge limited current, $\varepsilon_0$ is the vacuum permittivity ($8.85 \times 10^{-12}$ F m⁻¹), $\varepsilon_r$ is the permittivity for active layers, $\mu$ is the charge mobility, and $d$ is the thickness of active layers. The effective voltage ($V$) was obtained by the equation $V = V_{appl} - V_{bi} - V_s$, where the built-in voltage ($V_{bi}$) and the voltage drop ($V_s$) were deducted from the applied voltage ($V_{appl}$) of devices. Hole and electron mobilities were integrated from the slope of the $J^{1/2}$ - $V$ curves.

**Instruments and characterizations**

Ultraviolet−visible-near infrared (UV-vis-NIR) absorption was measured with the SHIMADZU UV-3600 spectrophotometer. Photoluminescence (PL) was acquired with a FM-4(HORIBA) spectrofluorimeter by a xenon arc lamp (150 W). Photoemission yield spectroscopy was performed with the model of AC3. The $J-V$ curves were measured under a computer controlled Keithley 2400 source meter under 100 mW cm⁻², AM 1.5 G solar simulator (Enlitech SS-F5). The $J-V$ curves were measured with forward scan mode from −0.2 V to 1.2 V, and the scan step was 0.02 V with dwell time of 1 ms. The light intensity was calibrated by a standard silicon solar cell

(certified by NREL) to give a range from 0.99 to 1.01 sun. Dark $J-V$ curves also obtained from the computer controlled Keithley 2400 source meter under dark. The light intensity dependence plot was obtained from the bult-in shutter of solar simulator with the intensity ranged from 10% to 100%. The EQE spectra were received from a commercial QE measurement system (Taiwan, Enlitech, QE-R3011). Hole and electron-only devices were recorded with a Keithley 236 sourcemeter under a dark chamber. The $EL_{EQE}$ was received from the emitted photons of devices in all directions through an integrated sphere via the QE Pro, Ocean Optics calibrated spectrometer (QE Pro, Ocean Optics), with various current density adjusted with a Keithley 2400 source meter. Tapping-mode AFM images were obtained by a Bruker Multimode 8 Microscope. TEM images were received from the JEM 2100 F Microscope. Cyclic voltammetry (CV) measurement was performed on CHI 630E Electrochemical workstation. The tetrabutylammonium hexafluorophosphate ($TBAPF_6$) was dissolved in acetonitrile with the concentration of 0.1 M. Nitrogen was ventilated into $TBAPF_6$ solution to remove oxygen. The saturated calomel electrode, platinum wire and glass carbon worked as reference electrode, counter electrode and working electrode, respectively. For the transient photocurrent (TPC) and the transient photovoltage (TPV) measurements, photogenerated charge carriers were excited at 580 nm by a mode-locked Ti:sapphire oscillator (SpectraPhysics Spitfire Ace) with a pulse width of 120 fs and the repetition rate of 1 kHz. TPC signal was recorded with an oscilloscope (Tektronix TDS 3052 C) with a 50 Ω resistor. TPV signal was recorded on a 1 MΩ resistor, under an LED light. Differential scanning calorimetry (DSC) were performed on DSC analyzer (DSC 200 F3, NETZSCH Scientific Instruments) with a heating/cooling rate of 10 K min⁻¹ from 30 to 330 °C, under a nitrogen atmosphere. Free standing films were taken off the glass substrates and samples of approximately 2.5 mg were filled into the DSC Al pan with pierced lid. Impedance spectra, $C-V$ and $C-F$ curves were obtained by KEYSIGHT impedance analyzer E4990A (20 Hz-10 MHz) in the dark, at room temperature. The $R_0$ and $R_1$ was obtained from the instrument in-built equivalent circuits.

**Capacitance–voltage ($C - V$) and capacitance-frequency ($C - \omega$)**
The capacitance–voltage ($C - V$) measurement and Mott-Shockley analysis can be used to explore the trap density ($N_A$) in OSCs. A linear region can be acquired from the Mott–Shockley plot ($C^{-2} - V$) whose intercept to the voltage axis represents the built-in voltage ($V_{bi}$). The trap density ($N_A$) is defined by the following equation of $N_A = \frac{-2}{q\varepsilon_r\varepsilon_0 A^2}\left(\frac{dV}{dC^{-2}}\right)$, where $\varepsilon_r$ is the relative dielectric constant, $\varepsilon_0$ is the vacuum permittivity, and $A$ is the effective area of devices. The electronic density of states (DOS) is obtained from capacitance-frequency ($C - \omega$) measurements under dark. The trap energy ($E_\omega$) can be expressed as a function of the applied frequency by $E_\omega = kT \ln(\frac{\omega_0}{\omega})$, where $k$ is Boltzmann's constant, $T$ is the temperature, and $\omega_0$ is the rate pre-factor ($\sim 10^{12}\,s^{-1}$) for thermal excitation from the trap in typical organic photodiodes. The trap DOS distribution is obeyed with $DOS_{(E_w)} = -\frac{\beta}{qAd}\frac{V_{bi}}{kT}\frac{\omega dC}{d\omega}$, where $V_{bi}$ is the built-in potential, $q$ is the electric charge, and $\beta$ is the correction factor equal to 1 in our analysis. Here, a widespread Gaussian shape was fitted with the equation of $DOS_{(E_w)} = \frac{N_t}{\sqrt{2\pi}\delta}\exp[-\frac{(E_t - E_w)^2}{2\delta^2}]$, where $N_t$ is the total density, $E_t$ is the center of the DOS, and $\delta$ is the disorder parameter.

**GIWAXS and RSoXS**
All the GIWAXS signals were recorded in Helium atmosphere using a 2D charge-coupled device (CCD) detector (Pilatus 2 M) with a pixel size of 0.172 mm by 0.172 mm. Thin film samples were coated atop PEDOT:PSS/silicon wafer substrates. The conditions for the BHJ solutions preparation and film treatment are the same as those for device fabrication. RSoXS was performed at beamline 11.0.1.2 of Advanced Light Source, LBNL[59]. Sample preparation was the same as that of GIWAXS samples. The BHJ films were separated via floating in deionized water and transferred on silicon nitrile windows, and subsequently the samples were loaded onto a holder. After the films were dried in air, the holder was transferred into the vacuum chamber at the beamline end station. The beam energy was screened ranging from 280 to 290 eV, with a 5 s exposure time per scan. The scattering patterns were collected in vacuum using Princeton Instrument PI-MTE CCD camera with a pixel size of 0.027 mm by 0.027 mm.

**Method of spectral decomposition**
Semi-quantitative structural features related to the morphological information can be extracted from UV-vis-NIR and PL spectra. For organic semi-conductors, ordered regions have increased π-π interactions with neighboring chains causing reduced torsional freedom. Moreover, translational periodicity allows for extended exciton wavefunctions. Therefore, ordered regions lead to red-shifted and structured aggregate bands in UV-vis-NIR and PL spectra (black dashed lines with shaded areas in Supplementary Fig. 27 for $A_0$ and red one in Supplementary Fig. 28 for $PL_0$, respectively). For the interchain interactions with stronger torsional mobility, these regions ascribed to amorphous bands, which are blue-shifted and less structured compared to the aggregate bands (red dashed lines with shaded areas in Supplementary Fig. 27 for $A_A$ and blue one in Supplementary Fig. 28 for $PL_A$, respectively). The noticeable $PL_A$ intensity can be modeled by a single Lorentz fitting. Assuming similar PL quantum yields ($\varnothing_{PL}$) in the ordered and amorphous phases, we can obtain a relative energy transfer quantum yield from the amorphous to the ordered phase with $\frac{\varnothing_{RET}}{\varnothing_{PL}} = (1 - \frac{PL_A}{PL_0}\frac{A_0}{A_a})/(\frac{PL_A}{PL_0} + 1)$. From the relative resonance energy transfer (RET) quantum yield, the amount of clustered amorphous phase and thus the ratio of ordered to amorphous range can estimated. Note that the emission of the amorphous phase occurs in the region of strong absorption and therefore will be strongly affected by reabsorption; hence, we can only obtain the approximate value of $\varnothing_{RET}/\varnothing_{PL}$. However, it still allowed us to unambiguously calculate the approximate structural features.

**Time-of-flight secondary ion mass spectrometry (ToF-SIMS)**
Depth profiles of the bilayer samples were measured with a TOF-SIMS 5-100 instrument (ION-TOF GmbH, Germany). The instrument was equipped with a dual beam mode comprising of a Bi/Mn liquid metal ion gun (LMIG) and an argon gas cluster ion gun. A 5 keV Ar-cluster beam was used for depth profiling by sputtering through the film in 5 s intervals over an area of 300 μm × 300 μm. The central area within the sputtered region was analyzed using a 30 keV Bi$^{3+}$ beam over an area of 100 μm × 100 μm. The obtained negative ion data was used for analysis.

The preparation of the bilayer samples involved several steps. First, solutions of PTzBI-dF (10 mg mL$^{-1}$), L8BO (14 mg mL$^{-1}$), and Y6 (14 mg mL$^{-1}$) in CF were separately prepared. NFAs was spin cast on pre-cleaned ZnO coated Si substrates at a spin-rate of 850 r min$^{-1}$. Neat PTzBI-dF film was spin cast on PEDOT:PSS coated glass at a spin-rate of 1000 r min$^{-1}$. Next, the PTzBI-dF film was then floated on DI water and transferred onto the NFAs/ZnO/wafer substrates, resulting in PTzBI-dF/NFA/ZnO/wafer bilayer samples. The samples were dried and can be post-thermally annealed at 100 °C for 5 min if necessary.

**Reporting summary**
Further information on research design is available in the Nature Portfolio Reporting Summary linked to this article.

## Data availability
Relevant data supporting the key findings of this study are available within the article and the Supplementary Information file. All raw data generated during the current study are available from the corresponding authors upon request.

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

## Acknowledgements

This work is financially supported by National Key Research and Development Program of China (Nos. 2022YFB4200400 and 2019YFA0705900), Guangdong Basic and Applied Basic Research Foundation (2023B1515040026), and Guangdong Major Project of Basic and Applied Basic Research (No. 2019B030302007). N.L. acknowledges the financial support by State Key Lab of Luminescent Materials and Devices, South China University of Technology (Skllmd-2022-03). W. Z. was supported by National Natural Science Foundation of China (Grant No. 22109094) and the fellowship of China Postdoctoral Science Foundation (No. 2022M712054). GIWAXS and RSoXS were performed at beamlines 7.3.3 and 11.0.1.2 at the Advanced Light Source, a U.S. DOE Office of Science User Facility under contract no. DE-AC02-05CH11231.

## Author contributions

K.A. performed the device characterization and analyzed the data, conducted the UV–vis-NIR, PL, DSC, J-V, EQE, stability measurements, C-V, C-W, Jph-Veff, photoemission yield spectroscopy, Plight versus JSC and VOC plots, hole-only and electron-only devices, AFM, TEM, impedance spectra, TPC and TPV measurements. W.Z., C.W., and F.L. performed the GIWAXS and RSoXS measurements and analyzed the data. F.P. synthesized materials. W.D. and H.W. performed the energy loss measurements and analyzed the data. Y.S. contributed to the scheme picture and helped with constructing the model of active layers decay. H.L.Q. helped with the C-V and C-W measurements. H.Q. performed the preparation of ZnO buffer layer and helped with thermal stability tests. N.L., F.H., L.Y. conceived the ideas and organized the work. K.A., N.L. and L.Y. wrote the manuscript with input from co-authors. All authors have commented on this manuscript.

## Competing interests

The authors declare no competing interests.
