## [Peer Review File · Nature Communications]

Mastering Morphology of Non-fullerene Acceptors towards Long-term Stable Organic Solar CellsREVIEWER COMMENTS

Reviewer #1 (Remarks to the Author):

In this article, An et al utilized the L8BO, Y6 and PTzBI-dF to constitute a ternary solar cells with high efficiency and robust morphology, and systematically investigated the reasons behind for the thermal stability of the ternary solar cells. Although, the PCE is not so attractive, however, the investigations on the morphology stability are of great significance, and the thermal stability is a key issue in the filed of OSCs. The manuscript is well organized, and the stable crystalline region and amorphous region of the ternary blends is described detailedly, using the binary solar cells as reference. I think the manuscript could be accepted by Nature Communications after the following major revisions:

1: In fact, the inverted device structure (ZnO/active layer/MoOx) is much more heating-stable than the conventional device (PEDOT:PSS/active layer/PF3N-Br/Ag). However, the performance in conventional device is usually much better than the inverted ones in the Y-series NFA systems. The device performance of the inverted structure should be shown in the Supporting Information, which is convenient for later stability data statistics. And this sentence in the abstract is misleading, which needs to be described clearly. "The PTzBI-dF:L8BO:Y6-based OSCs exhibit a promising PCE of 18.26% along with excellent long-term thermal stability over 1400 h under 85 °C continuous heating." Second, what is the trend for their thermal stability in the conventional devices? If it is not the same with inverted one, why?

2: The ΔH_{mix} variation rather than ΔH seems more reasonable to illustrate the improved stability. Because, without crystallization temperature in DSC, only applying ΔH to evaluate the crystalline ability seems not so persuadable. Furthermore, from your analysis below, the crystallinity in the ternary blend is not decreased as compared to the PTzBI-dF / L8BO. Additionally, the addition of polymer donor made the ΔH of the blends decrease 1.5-3 times, which is a large reduction, whether this big change is partly due to the nearly half weight ratio of the acceptors in the blend? From the suppressed ΔH or ΔH_{mix} in the blend, it seems PTzBI-dF / L8BO is more stable than PTzBI-dF / Y6. However, their stabilities have no big differences (Figure 2), why?

3: In the fullerene system, the glass forming alloy of fullerenes is reported to improve the stability (Adv Energy Mater, 2018, 8, 1702741) due to the increase in entropy upon mixing which would prohibit the phase separation, and it is also a universal approach to improve the device performances and also the stability. So, my questions are follows: first, for the similar structure of Y6 and L8BO, whether they form alloy or not? Second, from analysis of the TM and ΔH_{mix} , it seems the group of Y6 and L8BO is a robust combination, so what is about the stability improvement in other systems such as D18, PM6 and PTQ-10.

4: The variations of amorphous clusters in the fresh devices and aged devices in this article is described in detail. As the author illustrated the amorphous clusters would inhibit the efficient RET from the amorphous to the ordered region, leading to excitons dead before dissociation (according to the conservation of the mass, apparently it would also decrease the interfacial areas) and heavier charge recombination due to the trap accumulation. However, why it only makes big effect on FF but much less effect on the J_{sc} ? Furthermore, I am not persuaded by the calculation of Φ_{RET} / Φ_{PL} , although it may be reasonable. One reason is that the absorption spectrum and PL spectrum are quite complex, and the vibrational transitions in the amorphous states and aggregates can be largely overlapped. The other reason is that there is non-negligible existence of H aggregates in the NFAs. So, can you convince me this calculation could be applied in your system? Additionally, some other characterizations need to further support this idea. If the amorphous cluster changes to bigger ones after ageing, domain size would have differences which could be further supported by R-SoXs and Pi-FM.

5: Some small errors or suggestions. a) The number of Figures and Tables need to be carefully checked. For examples: Figure 4c should be Figure 3c, black dashed lines with shaded areas in Figure S15 for A0 and red one in Figure S16 for PLO, which should be Figure S17 and S18, respectively. b) Some sentences and statements should be described more accurately and consistently. For example, an important viewpoint in the article seems conflicted. "The addition of Y6 led to narrowed and intensified π - π stacking reflections in PTzBI-dF:L8BO:Y6 films, indicating improved crystallinity" "Thermodynamic regulation reveals that the melting endotherm (ΔH) can be minimized through constructing two NFAs with different thermodynamic responses to polymer donors, which significantly decreased crystallization behaviors of PTzBI271 dF:L8BO:Y6 compared to the PTzBI-dF:L8BO and PTzBI-dF:Y6 blends." c) The author states that there exists energy transfer between L8BO and Y6. For the small disparity in the intensity which could be due to the different thickness of the films and also the reduction of aggregation quench in Y6 because of its dilution by L8BO. I suggest if you think there is energy transfer between L8BO and Y6, more precise experiments are needed to design to prove it. d) The detailed characterization method on the measurement of Impedance spectroscopy should be included.

Reviewer #2 (Remarks to the Author):

In this manuscript, the authors studied the long-term stability of OSCs by integrating a wide bandgap electron-donating polymer and two NFAs. Devices based on PTzBI-Df:L8BO:Y6 exhibited a promising PCE of 18.26% and retained over 95% of its original PCE after being thermally annealed for about 1400 h. To select the two NFAs, the authors choose Y6 and L8BO, which has similar chemical structures but distinct thermodynamic properties. As a result, when the weight ratio of Y6 and L8BO is varied, the

thermodynamic properties of the active layer are tuned. And this is the critical factor to gaining better device stability. However, the amount of amorphous/crystalline phase is derived from the fitting to the absorbance spectra to support the critical statement, which is not convincing. Though the manuscript is well written with good readability, it should not be accepted at current stage. Additionally, some concerns should be addressed.

1. As shown in the supporting information S Fig 4d-f, the donor polymer PTzBI-Df does not have crystalline enthalpy. Therefore, quite similar in the L8BO and Y6 binary and ternary blends. When the ratio of L8BO/Y6 is 0.5:0.7 and 0.7:0.5, the thermal properties are more dominated by PTzBI-Df. As presented by Prof. Harald Ade, the molecular interaction-diffusion framework determines the organic solar cell stability (Nature Materials, 2021,20, 525–532). Therefore, could the readers understand that the amorphous feature is mainly due to the donor instead of the two NFAs? Because the weight ratio of the donor is the highest under these conditions.

2. The amorphous/crystalline phase in the active layer is critical to maintaining the device stability. However, the analysis of this is based on optical profiles. Though the absorbance spectra shape /intensity/band position is related to the thin film microstructure, it is not direct structural evidence. In addition, the authors cited references reported by Prof. Richard Friend and Deter Neher, which was studies on the rr-P3HT. It is well known that the rr-P3HT is a strong crystalline material, and the absorbance changes at lower energy upon post-treatment are directly relate to the crystalline/amorphous phase changes. The acceptor paired with rr-P3HT was often fullerene derivatives; the aggregation features differ from the NFAs. Therefore, their analysis or the correlation between absorbance and device performance could not be directly used in the NFAs compositions. Thus, the fittings of the absorbance/PL may mathematically fit, but whether it follows any physical properties remain debatable.

3. In the thermal stability part, it is found that the JSC and FF are not monotonously reduced upon thermal annealing, especially the ternary device and the L8BO binary device. The JSC has increased after about 500 h till 1000 h. The FF of the L8BO binary device has similar behavior. What could be the reason? Why do these parameters again go higher?

4. The statement of energy transfer between L8BO and Y6 is unclear to the readers. The spectral overlap between the crystalline/amorphous phase is fitted instead of directly measured. Therefore, the overlapping extent is questionable. On the other side, it is difficult to understand the RET from the amorphous cluster to the crystalline phase. During the measurement of photoluminescence, the excitation wavelength could excite both L8BO and Y6 and the PL spectra of L8BO:Y6 is the superposition of PL spectra of pristine L8BO and Y6. Therefore, it may not be proper evidence of RET. On page 10, the meanings of Φ_{RET} and Φ_{PL} are not clear.

5. Capacitance-voltage/frequency measurements determine the trap density, and the density of states (DOS) vs. trap energy are displayed in figure 4g, 4h, and 4i. As shown in figure 4, compared with PTzBI-Df:L8BO and PTzBI-Df:Y6, the DOS center position of PTzBI-Df:L8BO:Y6 moves towards higher energy. What could be the possible reasons?

Reviewer #3 (Remarks to the Author):

The manuscript “Mastering Microstructure Morphology of Non-fullerene Acceptors towards Long-term Stable Organic Solar Cells” by An et al. presented stable organic solar cells with high performances through a ternary blend strategy. Such phenomenon is observed in various reports (Nat. Mater. 2017, 16, 363; Adv. Mater. 2019, 1904601; ACS AMI, 2017, 9, 24, 20704; J. Mater. Chem. C. 2022, 10, 3207) with thermodynamic origins established (Adv. Energy Mater. 2018, 1702741). The authors employed diffraction, photonic spectroscopy, and thermal analysis to establish the nanoscale structure modification enabled by the mixing of two similar acceptor molecules. However, these analyses did not support the nanostructure illustrated by the authors due to either inappropriate application of the models or misinterpretation of the data. The main arguments of the manuscript including “better crystalline quality”, “attained thermodynamic equilibrium”, “suppress of amorphous aggregate to cluster” and “different thermodynamic diffusion behaviors” were not supported by the experimental results presented in the manuscript. Hence, I cannot support the publication of this manuscript in Nature Communications. Here are my comments in detail.

1. The authors named this manuscript “mastering microstructure ... organic solar cells”. In fact, the morphology of organic solar cells matters in nanoscale. The term “microstructure” refers to structures in micrometer scale. The author should revise the manuscript for this error.
2. The authors discussed the gradual evolution of morphology of BHJ to the equilibrium state without considering the glass transition of the system, while there are numerous studies showed high glass transition can suppress such evolution. The authors should provide related discussions.
3. Most processing procedures for OSC undergo a prolonged thermal annealing process at or above 100 °C, which is significantly higher than the service temperature of the OSCs. It is not suitable for stating “post treatments ... result in non-equilibrium morphology”. Furthermore, a true equilibrium morphology for a two-phase structure will be either a double layered system (i.e. oil on water, for immiscible) or solid solution (i.e. metal alloys, for miscible). It is against the fundamental requirements of OSC for pursuing equilibrium morphologies which cannot be BHJ. Can the authors clarify how the “attained thermodynamic equilibrium” is established in this work and how a molecularly mixed or two layered equilibrium structure providing such high performances?
4. The work is largely based on morphology study through spectroscopy. However, all the theoretical models of photonic spectroscopy established by numerous work of F. Spano, C. Silva, J. Clark and D. Neher are all based on P3HT which is a flexible molecule. The typical polymers used in current top performing OSCs are significantly more rigid (Nano Letters 2013, 13, 6, 2522) where models describing the photonic spectroscopy properties are yet established. The high throughput references the authors cited are based on matching device performances with spectrum of specific materials with theoretical models not being established. Hence, the authors cannot give such specific assignment to the “PL at NIR region” to “the clustered amorphous phase” without theoretical models established on specific materials used in this work or based on models established for P3HT which is a very different material. The discussions the authors offered on the spectroscopy measurements can only support “morphology

can be fine-tuned by changing the blend ratio of NFAs”, but not “The amorphous phases in the PTzBI:L8BO:Y6 film could be effectively prohibited into cluster under thermal stress, which occurred in PTzBI:L8BO and PTzBI:Y6 films probed by optical spectroscopy” as in the discussion and introduction, and certainly not the images provided in Scheme 1 and S16.

5. The authors referred L8BO and Y6 with “different thermodynamic properties”. Do the author mean thermodynamic properties such as heat capacitance and formation entropy? Can the authors provide specific values?

6. Please provide the line cuts for the blend GIWAXS images and a demonstration of the fitting same as the spectroscopy analyses. The authors deduced “better crystalline quality” based on higher CCL values of pi-pi stacking for Y6 over L8BO. What do the authors mean by “better crystalline quality” do the author mean higher crystallinity or lower percentage of dislocations? Please provide values if available. Also the CCL values quoted in the main text is not consistent with Table S2.

7. The authors compared the CCL of the pi-pi stacking L8BO and Y6. However, based on the single crystal structures of the two materials (Nature Energy 2021, 6, 605), the pi-pi stacking of Y6 containing two packing forms and L8BO containing three packing motifs. Hence, the resulting peak around 1.7 is the superposition of several peaks which is supported by the similar CCL calculated from the in-plane scatterings while such large CCL differences out-of-plane. Furthermore, such similarity of crystal sizes is supported by the RSoXS data where identical small angle scattering angle was found for all thin films. Hence, it is not valid to estimate CCL from the pi-pi stacking using FWHM obtained in scatterings containing more than one peaks in the case of this manuscript.

8. The thermal analysis presented in this work displayed a very interesting eutectic formation (Advanced Materials 2008, 20, 18, 3510). Hence, the thermal behavior of the blends cannot be simply compared with the enthalpy of fusion as the eutectic phase normally presents lower enthalpy of fusion for the presence of non-negligible surface energy. The lower enthalpy of fusion cannot be considered as the prove for “exhibit most robust morphological stability under thermal stress”. Furthermore, “L8BO and Y6 show different thermodynamic diffusion behaviors into PTzBI-dF” cannot be assessed by the thermal analysis. I suggest the authors to prepare bilayer structures and perform diffusion analysis using TOF-SIMS method to support this argument. (Macromolecules 2013, 46, 3, 1002) The phase behavior of similar acceptors blend is explored by Baran group with similar results, which I suggest the authors to go through. (ACS Energy Letter 2020, 5, 5, 1371; Advanced Functional Materials 2020, 2005462)

9. The authors should clarify how RSoXS data with almost identical scattering position can support “blending L8BO and Y6 can regulate the crystallinity and phase-separated length scale of BHJ thin films, opening an avenue to fine-tune nanostructure and photophysical properties of these blends.”

10. The author found a “significantly decreased crystallization behaviors of PTzBI-dF:L8BO:Y6 compared to the PTzBI-dF:L8BO and PTzBI-dF:Y6 blends.” The significantly decreased crystallization should be detrimental to the charge transport performance. However, they PTzBI-dF:L8BO:Y6 exhibited higher charge carrier mobilities than the binary blends. The authors should provide reasonable explanations on this.

REVIEWER COMMENTS

Reviewer #1 (Remarks to the Author):

In this article, An et al utilized the L8BO, Y6 and PTzBI-dF to constitute a ternary solar cells with high efficiency and robust morphology, and systematically investigated the reasons behind for the thermal stability of the ternary solar cells. Although, the PCE is not so attractive, however, the investigations on the morphology stability are of great significance, and the thermal stability is a key issue in the filed of OSCs. The manuscript is well organized, and the stable crystalline region and amorphous region of the ternary blends is described detailly, using the binary solar cells as reference. I think the manuscript could be accepted by Nature Communications after the following major revisions:

1: In fact, the inverted device structure (ZnO/active layer/MoOx) is much more heating-stable than the conventional device (PEDOT:PSS/active layer/PF3N-Br/Ag). However, the performance in conventional device is usually much better than the inverted ones in the Y-series NFA systems. The device performance of the inverted structure should be shown in the Supporting Information, which is convenient for later stability data statistics. And this sentence in the abstract is misleading, which needs to be described clearly. “The PTzBI-dF:L8BO:Y6-based OSCs exhibit a promising PCE of 18.26% along with excellent long-term thermal stability over 1400 h under 85 °C continuous heating.” Second, what is the trend for their thermal stability in the conventional devices? If it is not the same with inverted one, why?

Our response: We are grateful to the referee for the very helpful comments. In the revised manuscript, we have added the photovoltaic performance of inverted OSCs in Supplementary Fig. 17 and Supplementary Table 9. Relevant sentences were added in the revised manuscript on Page 8, 9.

Page 8, 9: “The inverted structure has been proven to prevent the potential degeneration factors from interfaces [*Joule*, 2019, 3, 215-226; *Adv. Mater.*, 2020, 32, 1908305]. The photovoltaic performance of PTzBI-dF:L8BO, PTzBI-dF:Y6 and PTzBI-dF:L8BO:Y6-based inverted devices lags behind those of the conventional structure counterparts (Supplementary Fig. 17, Supplementary Table 9). However, slightly higher efficiencies can still be achieved in PTzBI-dF:L8BO:Y6-based devices.”

To avoid misleading, we replaced the original description in “Abstract” with the following sentences.

Abstract: “The OSCs based on PTzBI-dF:L8BO:Y6 could achieve a promising PCE of 18.26% in the conventional device structure. In the inverted structure, excellent long-term thermal stability over 1400 h under 85 °C continuous heating is obtained.”

Supplementary Fig. 17 | Photovoltaic performance. The J - V characteristics of PTzBI-dF:L8BO, PTzBI-dF:Y6 and PTzBI-dF:L8BO:Y6 inverted devices under simulated AM 1.5G irradiation (100 mW cm^{-2}).

Supplementary Table 9 | Photovoltaic parameters of PTzBI-dF:L8BO, PTzBI-dF:Y6, PTzBI-dF:L8BO:Y6 inverted devices under simulated AM 1.5G irradiation (100 mW cm^{-2})

Active layer	V_{oc} (V)	J_{sc} (mA cm^{-2})	FF (%)	PCE (%)
PTzBI-dF:L8BO	0.897	23.36	72.29	15.15
PTzBI-dF:Y6	0.840	25.25	70.72	14.92
PTzBI-dF:L8BO:Y6	0.876	25.12	73.83	16.25

As suggested by the reviewer, the thermal stability of PTzBI-dF-based OSCs with conventional structure were evaluated. As depicted in Supplementary Fig. 16, all conventional devices encountered similarly undesirable degradation under continuous $85 \text{ }^\circ\text{C}$ thermal stress, indicating that buffer layer degeneration plays a critical role to the instability of OSCs. PEDOT:PSS interface possesses the hygroscopicity, acidity, and anisotropic charge injection properties, which could aroused the intrinsic instability to OSCs [*Sol. Energy Mater. Sol. Cells*, 2008, 92, 686–714; *Energy Environ. Sci.*, 2012, 5, 6521; *Phys. Chem. Chem. Phys.*, 2011, 13, 4381–4387].

To clarify this issue, we have amended relevant sentences in Page 8 and Supplementary Fig. 16 in the revised manuscript.

Page 8: “All conventional devices encountered similarly undesirable degradation to less than 80% within 200 h, indicating the intrinsic instability caused by PEDOT:PSS buffer layers (Supplementary Fig. 16).”

Supplementary Fig. 16 | Thermal stability. The thermal stability of PTzBI-dF:L8BO, PTzBI-dF:Y6, PTzBI-dF:L8BO:Y6 devices with conventional structure under continuous 85 °C ageing in a dry nitrogen atmosphere without encapsulation.

2: The ΔH_{mix} variation rather than ΔH seems more reasonable to illustrate the improved stability. Because, without crystallization temperature in DSC, only applying ΔH to evaluate the crystalline ability seems not so persuadable. Furthermore, from your analysis below, the crystallinity in the ternary blend is not decreased as compared to the PTzBI-dF / L8BO. Additionally, the addition of polymer donor made the ΔH of the blends decrease 1.5-3 times, which is a large reduction, whether this big change is partly due to the nearly half weight ratio of the acceptors in the blend? From the suppressed ΔH or ΔH_{mix} in the blend, it seems PTzBI-dF / L8BO is more stable than PTzBI-dF / Y6. However, their stabilities have no big differences (Figure 2), why?

Our response: We greatly thank the referee for the helpful comment. According to the suggestion, we calculated the ΔH_{mix} variation from ΔH via deducting the weight of PTzBI-dF (Supplementary Fig. 6).

There also existed inaccurate narrative for the relationship between “suppressed ΔH or ΔH_{mix} ” and “the crystalline ability”, as pointed out by the referee. For our current understanding, the use of fullerene mixtures tends to impede the tendency for crystallization of the multicomponent blend, reported by previous work [*Adv. Mater.*, 2015, 27, 7325; *Adv. Mater.*, 2016, 28, 8021]. However, NFAs tend to crystallize under external stress. Mixing several components of NFAs has met with limited success to impede the crystallization tendency due to the complex nanostructure of the resulting multicomponent blends [*Nat. Rev. Mater.*, 2019, 4, 229-242]. As a result, mixing of several NFAs is not a guarantee that crystallization tendency can be suppressed. The

decrease in enthalpy upon mixing NFAs may be related to the form of a homogeneous amorphous mixture in similar acceptors blend, according to the previous reports [*Adv. Funct. Mater.*, 2020, 30, 2005462; *ACS Energy Lett.*, 2020, 5, 1371–1379]. In the revised manuscript, we removed relevant narrative for the relationship between “suppressed ΔH or ΔH_{mix} ” and “the crystalline ability”. The ΔH or ΔH_{mix} was only proposed a guiding criterion to select representative PTzBI-dF:L8BO:Y6 blends to evaluate the thermal stability of active layers, as the a well-mixed amorphous solid phase may occurred.

The value of ΔH cannot be directly used to determine the devices stability. Hence, we also de-emphasize the narrative about the relationship between “lower enthalpy” and “robust morphological stability”. Generally, we have specified or de-emphasize some narrative in Page 6 and added Supplementary Fig. 6 in the revised manuscript. We highly appreciate the reviewer’s insightful comments.

Page 6: “The T_{m2} related to Y6 pinned at about 280 °C, suggesting the melting point could be easily affected even mixed with small amounts of L8BO (Supplementary Fig. 5c). ... but PTzBI-dF:L8BO:Y6 blends exhibited a broadened melting transition due to the NFAs. ... The ΔH_{mix} exhibits most suppress with 6.07 J g⁻¹ in 1:0.7:0.5 blend, indicating that the ΔH can be minimized through constructing two NFAs with different thermal proprieties (Supplementary Fig. 6). ... We focus on the blend ratio of 1:0.7:0.5 PTzBI-dF:L8BO:Y6 blends, as the mixed NFAs with similar structure may form a homogeneous amorphous mixture in blends, which also offers the best photovoltaic performance (cf. discussion on device performance below).”

Supplementary Fig. 6 | Thermal behavior. The ΔH_{mix} variation obtained from the DSC of PTzBI-dF:L8BO:Y6 blends via deducting the weight of the polymer donor PTzBI-dF.

3: In the fullerene system, the glass forming alloy of fullerenes is reported to improve

the stability (Adv Energy Mater, 2018, 8, 1702741) due to the increase in entropy upon mixing which would prohibit the phase separation, and it is also a universal approach to improve the device performances and also the stability. So, my questions are follows: first, for the similar structure of Y6 and L8BO, whether they form alloy or not? Second, from analysis of the TM and ΔH_{mix} , it seems the group of Y6 and L8BO is a robust combination, so what is about the stability improvement in other systems such as D18, PM6 and PTQ-10.

Our response: We greatly thank the referee for the very helpful comment. Acceptor alloys can be formed in L8BO:Y6, considering the similar structure of L8BO and Y6 acceptors. The cyclic voltammetry (CV) measurement was used to further evaluate the energy level of L8BO:Y6. To clarify this issue, we have added relevant sentences in Page 9, Instruments and characterizations section and Supplementary Fig. 19 of the revised manuscript.

Page 9: “Acceptor alloys could be formed in L8BO:Y6, because the similar structure of L8BO and Y6, and a linear dependence of V_{OC} varied with the stoichiometry of L8BO and Y6 in OSCs (Supplementary Table 5). The cyclic voltammetry (CV) measurement was used to further determine the energy levels of L8BO:Y6. When L8BO and Y6 blended, the mixed acceptors show shifted energy levels dependent on the components (Supplementary Fig. 19, Supplementary Table 10), which reinforced the formation of effective alloy for L8BO:Y6 acceptors. [Adv. Energy Mater., 2018, 8, 1702741]”

Instruments and characterizations: “Cyclic voltammetry (CV) measurement was performed on CHI 630E Electrochemical workstation. The tetrabutylammonium hexafluorophosphate (TBAPF₆) was dissolved in acetonitrile with the concentration of 0.1 M. Nitrogen was ventilated into TBAPF₆ solution to remove oxygen. The saturated calomel electrode, platinum wire and glass carbon worked as reference electrode, counter electrode and working electrode, respectively.”

Supplementary Fig. 19 | Cyclic voltammetry. (a) Ionization energy and (b) electron affinity of L8BO, Y6 and L8BO:Y6 acceptors determined by cyclic voltammetry (CV). (c) Half-wave potential of ferrocene reference. (d) Derivative of corresponding absorption spectra.

Supplementary Table 10 | Parameters of energy levels and optical bandgap

Acceptors	E_{ox}^a (V)	HOMO (eV)	E_{re}^b (V)	LUMO (eV)	λ_{onset} (nm)	$E_{gap, optical}^c$ (eV)
L8BO	1.33	-5.73	-0.53	-3.87	836.9	1.48
Y6	1.28	-5.68	-0.50	-3.90	874.5	1.42
L8BO:Y6	1.30	-5.70	-0.51	-3.89	854.0	1.45

^a $E_{HOMO} = -[E_{ox} - E_{Fc/Fc+} + 4.8]$ eV; ^b $E_{LUMO} = -[E_{re} - E_{Fc/Fc+} + 4.8]$ eV; ^c Optical bandgap extracted from absorption spectra.

According to the reviewer's suggestion, we carried out additional experiments and employed D18, PM6 and PTQ-10 as alternative polymer donors to investigate the stability improvement in OSCs. The weight ratio of polymer donors:L8BO:Y6 was identical in our work without using additives or other post-treatments, to eliminate the complex effects on the active layer caused by different processing methods reported by previously work [*Adv. Mater.*, 2022, 34, 2204718; *Sci. Bull.*, 2022, 65, 272–275; *Joule*, 2019, 3, 1140–1151; *Nat. Energy*, 2021, 6, 605–613; *Adv. Mater.*, 2019, 31, 1905480]. We were not able to get a good PTQ-10 batch in the limited time, but the inverted

devices based on both D18 and PM6 polymer donors exhibited decent photovoltaic performance (Supplementary Fig. 20, Supplementary Table 11), which can be used to conduct long-term thermal stability test. As shown in Supplementary Fig. 21–22, L8BO:Y6 combinations led to improved thermal stability when combined with D18 and PM6 polymer donors than those for binary devices. However, all PTQ10-based devices showed poor stability (Response letter | Figure 1, 2), which may be related to the poor quality of the batch we have received [*ACS Appl. Mater. Interfaces*, 2019, 11, 18555–18563]. These promising stability results further verify the effectiveness of the acceptor alloys strategy demonstrated in our work. To clarify this issue, we have added relevant sentences in Page 9, Materials section, Device fabrication section, Supplementary Fig. 20-22 and Supplementary table 11 of the revised manuscript.

Page 9: “Commercially available polymer donors of D18 and PM6 were utilized to further investigate the stability improvement with the robust combination of L8BO:Y6 acceptors. The L8BO:Y6 combination led to improved thermal stability when combined with D18 and PM6 polymer donors than those for reference devices (Supplementary Fig. 21, 22). The degeneration of FF was effectively inhibited. The D18:L8BO:Y6 and PM6:L8BO:Y6 devices showed the decay of ~10% and ~20% of its original PCE after being thermally annealed for about 500 h, indicating a good improvement in thermal stability. Hence, it was a feasible strategy to utilize acceptor alloys strategy to improve the stability of OSCs.”

Materials: “PTzBI-dF, PM6, D18, L8BO, Y6 and PNDIT-F3N-Br were purchased from Volt-Amp Optoelectronics Tech. Co., Ltd, China.”

Device fabrication: “The PTzBI-dF-based active layers deposited as mentioned above. The used active layer of D18:NFA and PM6:NFA (1:1.2, wt:wt) were dissolved in chloroform (CF) with the total concentration of 11 and 15 mg mL⁻¹. The film thickness controlled at about 100-110 nm. The fabricated active layer was followed with thermal annealing at 100 °C for 5 min.”

Supplementary Fig. 20 | Photovoltaic performance. (a) The chemical structure of D18 and PM6 polymer donors. (b, c) The *J-V* curves of D18 and PM6 separately combined with L8BO, Y6 and L8BO:Y6 as active layers.

Response letter | Figure 1. (a) The chemical structure of PTQ10 polymer donor. (b) The *J-V* curves of PTQ10 separately combined with L8BO, Y6 and L8BO:Y6 as active layers.

Supplementary Fig. 21 | Thermal stability. The thermal stability of D18:L8BO, D18:Y6 and D18:L8BO:Y6 devices under continuous 85 °C ageing in a dry nitrogen atmosphere without encapsulation.

Supplementary Fig. 22 | Thermal stability. The thermal stability of PM6:L8BO, PM6:Y6 and PM6:L8BO:Y6 devices under continuous 85 °C ageing in a dry nitrogen atmosphere without encapsulation.

Response letter | Figure 2. The thermal stability of PTQ10:L8BO, PTQ10:Y6 and PTQ10:L8BO:Y6 devices under continuous 85 °C ageing in a dry nitrogen atmosphere without encapsulation

Supplementary Table 11 | Photovoltaic parameters of D18 and PM6 separately combined with L8BO, Y6 and L8BO:Y6 as active layers-based OSCs under simulated AM 1.5G irradiation (100 mW cm⁻²)

Active layers	V_{oc} (V)	J_{sc} (mA cm ⁻²)	FF (%)	PCE (%)
D18:L8BO	0.885	24.75	65.57	14.47
D18:Y6	0.827	25.49	69.60	14.80
D18:L8BO:Y6	0.854	24.71	72.15	15.35
PM6:L8BO	0.867	25.20	68.13	14.89
PM6:Y6	0.814	26.24	69.04	14.76
PM6:L8BO:Y6	0.855	25.39	68.40	14.97

Response letter | Table1. Photovoltaic parameters of PTQ10 separately combined with L8BO, Y6 and L8BO:Y6 as active layers based OSCs under simulated AM 1.5G irradiation (100 mW cm⁻²)

Active layers	V _{oc} (V)	J _{sc} (mA cm ⁻²)	FF (%)	PCE (%)
PTQ10:L8BO	0.867	23.24	60.14	12.13
PTQ10:Y6	0.821	23.66	66.94	13.01
PTQ10:L8BO:Y6	0.857	24.23	62.25	12.94

4: The variations of amorphous clusters in the fresh devices and aged devices in this article is described in detail. As the author illustrated the amorphous clusters would inhibit the efficient RET from the amorphous to the ordered region, leading to excitons dead before dissociation (according to the conservation of the mass, apparently it would also decrease the interfacial areas) and heavier charge recombination due to the trap accumulation. However, why it only makes big effect on FF but much less effect on the J_{sc}? Furthermore, I am not persuaded by the calculation of ϕ_{RET} / ϕ_{PL} , although it may be reasonable. One reason is that the absorption spectrum and PL spectrum are quite complex, and the vibrational transitions in the amorphous states and aggregates can be largely overlapped. The other reason is that there is non-negligible existence of H aggregates in the NFAs. So, can you convince me this calculation could be applied in your system? Additionally, some other characterizations need to further support this idea. If the amorphous cluster changes to bigger ones after ageing, domain size would have differences which could be further supported by R-SoXs and Pi-FM.

Our response: According to literature, excitons dead and heavier charge recombination due to the trap accumulation accounting the FF and J_{sc} losses are normally observed, especially for fullerene-based devices [*Nat. Commun.*, 2017, 8, 14541; *ACS Appl. Mater. Interfaces*, 2019, 11, 18555]. However, lots of works reported these electron traps lead to losses mainly in FF for NFAs. Semiquantitative expressions of FF have developed which quantifies the FF factors change so much with thickness, light intensity and materials properties [*Nat. Commun.*, 2015, 6, 7083]. Meanwhile, it has been established that in the mixed regions of NFAs based OSCs, due to reorganization of polymer chains and diffusion-limited aggregation of NFAs to assemble small isolated acceptor domains, broaden the density of states and reduce electron mobility under weak electric field, which in particular affects FF of OSCs [*ACS Appl. Mater. Interfaces*, 2019, 11, 21766–21774; *Adv. Mater.*, 2020, 32, 1908305]. To clarify this issue, we have added relevant sentences in Page 9 of the revised manuscript.

Page 9: “The FF exhibited the biggest drop, while the V_{oc} and J_{sc} remained

relatively robust, indicating degradation predominantly occurred in the mixed region.”

Optical spectrum is effective auxiliary measurements for film morphology. Even if there exists significant spectral crowding, we are still able to extract morphology related features relying on Frank-Condon transitions and weak H aggregates (the so-called “Spano” model) [*ChemSusChem*, 2021, 14, 3590-3598]. Optical spectrum has many advantages, such as highly efficient, convenient, strong operability, and can be combined with high-throughput methods, which has been utilized to different complex materials system. This method has been demonstrated to work well with NFAs, where morphology was found to be strongly correlated with electrical performance and degradation [*Joule*, 2021, 5, 495-506]. Moreover, recently work showed that morphological features can also be extracted from all-polymer systems [*Nat. Energy*, 2022, 7, 1180-1190]. It is known that some small molecule systems such as Y6 present more than one crystalline phase. Morphological information extracted from UV-vis-NIR and PL spectra in this manuscript are only semi-quantitative structural features. Therefore, we also evaluated our blends morphology with GIWAXS measurements identifying the ordered phases to support our conclusion. According to the reviewer’s suggestion, we conducted alternative TEM and RSoXS measurements for aged PTzBI-dF:L8BO, PTzBI-dF:Y6 and PTzBI-dF:L8BO:Y6 blends to support the amorphous aggregate in our films. To clarify this issue, we have added relevant sentences in Page 11 and Supplementary Fig. 30 of the revised manuscript.

Page 11: “TEM and RSoXS characterizations were performed to evaluate the domain change in aged blends. Both PTzBI-dF:L8BO and PTzBI-dF:Y6 blends exhibited enlarged distribution of bright and dark regimes as well as obviously changed humps within the probed q range (Supplementary Fig. 30), which supported large amorphous cluster in films observed in optical spectrum, combined with the nearly identical GIWAXS patterns.”

Supplementary Fig. 30 | Morphology. (a) TEM images and (b) RSoXS averaged profiles for fresh and aged PTzBI-dF:L8BO, PTzBI-dF:Y6 and PTzBI-dF:L8BO:Y6 films.

5: Some small errors or suggestions.

a) The number of Figures and Tables need to be carefully checked. For examples: Figure 4c should be Figure 3c, black dashed lines with shaded areas in Figure S15 for A0 and red one in Figure S16 for PL0, which should be Figure S17 and S18, respectively.

Our response: According to the referee’s suggestion, we have rechecked the number of Figures and Tables in the revised manuscript.

b) Some sentences and statements should be described more accurately and consistently. For example, an important viewpoint in the article seems conflicted. “The addition of Y6 led to narrowed and intensified π – π stacking reflections in PTzBI-dF:L8BO:Y6 films, indicating improved crystallinity” “Thermodynamic regulation reveals that the melting endotherm (ΔH) can be minimized through constructing two NFAs with different thermodynamic responses to polymer donors, which significantly decreased crystallization behaviors of PTzBI271 dF:L8BO:Y6 compared to the PTzBI-dF:L8BO and PTzBI-dF:Y6 blends.”

Our response: We greatly thank the referee for the helpful comment. As the reviewer suggested, we have modified some narrative in the revised manuscript.

Page 10: “The addition of Y6 led to narrowed and intensified π – π stacking reflections in PTzBI-dF:L8BO:Y6 films, indicating better molecular packing.”

Discussion: “The melting endotherm (ΔH) can be minimized through

constructing two NFAs possessing different thermal properties with polymer donors.”

c) The author states that there exists energy transfer between L8BO and Y6. For the small disparity in the intensity which could be due to the different thickness of the films and also the reduction of aggregation quench in Y6 because of its dilution by L8BO. I suggest if you think there is energy transfer between L8BO and Y6, more precise experiments are needed to design to prove it.

Our response: To avoid misleading, we have removed the relevant statement and Figure in the revised manuscript.

d) The detailed characterization method on the measurement of Impedance spectroscopy should be included.

Our response: Thanks for the comments. We have added detailed characterization method on the measurement of Impedance spectroscopy in Instruments and characterizations section.

Instruments and characterizations: “Impedance spectra, $C-V$ and $C-F$ curves were obtained by KEYSIGHT impedance analyzer E4990A (20 Hz-10 MHz) in the dark, at room temperature. The R_0 and R_1 was obtained from the instrument in-built equivalent circuits.”

Supplementary Fig. 31 | Equivalent circuits. The equivalent circuits used in impedance spectra.

Reviewer #2 (Remarks to the Author):

In this manuscript, the authors studied the long-term stability of OSCs by integrating a wide bandgap electron-donating polymer and two NFAs. Devices based on PTzBI-Df:L8BO:Y6 exhibited a promising PCE of 18.26% and retained over 95% of its original PCE after being thermally annealed for about 1400 h. To select the two NFAs, the authors choose Y6 and L8BO, which has similar chemical structures but distinct thermodynamic properties. As a result, when the weight ratio of Y6 and L8BO is varied, the thermodynamic properties of the active layer are tuned. And this is the critical factor to gaining better device stability. However, the amount of amorphous/crystalline phase is derived from the fitting to the absorbance spectra to support the critical statement, which is not convincing. Though the manuscript is well written with good readability, it should not be accepted at current stage. Additionally, some concerns should be addressed.

1. As shown in the supporting information S Fig 4d-f, the donor polymer PTzBI-Df does not have crystalline enthalpy. Therefore, quite similar in the L8BO and Y6 binary and ternary blends. When the ratio of L8BO/Y6 is 0.5:0.7 and 0.7:0.5, the thermal properties are more dominated by PTzBI-Df. As presented by Prof. Harald Ade, the molecular interaction-diffusion framework determines the organic solar cell stability (Nature Materials, 2021,20, 525–532). Therefore, could the readers understand that the amorphous feature is mainly due to the donor instead of the two NFAs? Because the weight ratio of the donor is the highest under these conditions.

Our response: We greatly thank the referee for the helpful comment. We admitted the amorphous feature is dominated by polymer PTzBI-dF. Without enthalpy of fusion of polymer, only applying ΔH originated from NFAs to evaluate the stability seems not so persuadable. In the revised manuscript, we removed relevant narrative for the relationship between “suppressed ΔH or ΔH_{mix} ” and “morphological stability”. The ΔH or ΔH_{mix} was only proposed a guiding criterion to select representative PTzBI-dF:L8BO:Y6 blends to evaluate the thermal stability of active layers, as the well-mixed amorphous solid phase may occurred [ACS Energy Lett., 2020, 5, 1371–1379; Adv. Funct. Mater., 2020, 30, 2005462].

Page 6: “We focus on the blend ratio of 1:0.7:0.5 PTzBI-dF:L8BO:Y6 blends, as the mixed NFAs with similar structure may form a homogeneous amorphous mixture in blends, which also offers the best photovoltaic performance (cf. discussion on device performance below).”

Page 6: The original narrative: “Compared to the pristine L8BO and Y6 films, the ΔH values of PTzBI-dF:L8BO and PTzBI-dF:Y6 are suppressed by factors of 3 and 1.5, respectively, suggesting that L8BO and Y6 show different thermodynamic diffusion

behaviors into PTzBI-dF.” and “The 1:0.7:0.5 blends may exhibit most robust morphological stability under thermal stress, as the minimized crystallization behaviors compared to the PTzBI-dF:L8BO and PTzBI-dF:Y6 blends.” was removed to avoid misleading.

2. The amorphous/crystalline phase in the active layer is critical to maintaining the device stability. However, the analysis of this is based on optical profiles. Though the absorbance spectra shape /intensity/band position is related to the thin film microstructure, it is not direct structural evidence. In addition, the authors cited references reported by Prof. Richard Friend and Deter Neher, which was studies on the rr-P3HT. It is well known that the rrP3HT is a strong crystalline material, and the absorbance changes at lower energy upon post-treatment are directly relate to the crystalline/amorphous phase changes. The acceptor paired with rr-P3HT was often fullerene derivatives; the aggregation features differ from the NFAs. Therefore, their analysis or the correlation between absorbance and device performance could not be directly used in the NFAs compositions. Thus, the fittings of the absorbance/PL may mathematically fit, but whether it follows any physical properties remain debatable.

Our response: We thank the referee for the comments. In fact, optical spectrum is an effective auxiliary measurement to film morphology. Relying on Frank-Condon transitions and weak H aggregates (the so-called “Spano” model), physics-aware extraction of morphological features from UV-vis-NIR and PL spectra are semi-quantitative structural features in this manuscript [*ChemSusChem*, 2021, 14, 3590-3598]. It allowed us to unambiguously calculate the approximate structural features. This method has been demonstrated to work well to NFAs, where morphology was found to be strongly correlated with electrical performance and degradation [*Joule*, 2021, 5, 495-506]. Moreover, recently work showed that morphological features can also be extracted from all-polymer systems [*Nat. Energy*, 2022, 7, 1180-1190]. In this manuscript, we want to use optical spectrum as a useful and convenient tool to evaluate our film morphology. We also evaluated our blends morphology with GIWAXS measurements, identifying the ordered phases to support our conclusion. In the revised manuscript, we conducted TEM and RSoXS measurements for aged PTzBI-dF:L8BO, PTzBI-dF:Y6 and PTzBI-dF:L8BO:Y6 blends to support the amorphous aggerate in our films (Supplementary Fig. 30). To clarify this issue, we have added relevant sentences in Page 11 and Supplementary Fig. 30 of the revised manuscript.

Page 11: “TEM and RSoXS characterizations were performed to evaluate the domain change in aged blends. Both PTzBI-dF:L8BO and PTzBI-dF:Y6 blends exhibited enlarged distribution of bright and dark regimes as well as obviously changed humps within the probed q range (Supplementary Fig. 30), which supported large

amorphous cluster in films observed in optical spectrum, combined with the nearly identical GIWAXS patterns.”

Supplementary Fig. 30 | Morphology. (a) TEM images and (b) RSoXS averaged profiles for fresh and aged PTzBI-dF:L8BO, PTzBI-dF:Y6 and PTzBI-dF:L8BO:Y6 films.

3. In the thermal stability part, it is found that the J_{SC} and FF are not monotonously reduced upon thermal annealing, especially the ternary device and the L8BO binary device. The J_{SC} has increased after about 500 h until 1000 h. The FF of the L8BO binary device has similar behavior. What could be the reason? Why do these parameters again go higher?

Our response: We greatly thank the referee for the comments. The long-term thermal stability test in this work was not performed with in-situ instruments. It was manual tests after every continuously heating time points. Due to test conditions, the photovoltaic performance of devices will inevitably fluctuate with the long heating time. Here, three represent testing plots of PTzBI-dF:L8BO and PTzBI-dF:L8BO:Y6-based devices ageing 0 h, 384 h and 552 h, separately were selected (Response letter Figure 3–5). The distribution and average values of photo-shunt and photovoltaic parameters extracted from J - V curves indeed showed inevitable fluctuation during the long-term heating process. However, each point was obtained from statistical average. The degradation tendency of our devices is reliable.

Response letter | Figure 3. (a, c) J - V characteristics and (b, d) corresponding shunt resistance (R_{sh}) for PTzBI-dF:L8BO and PTzBI-dF:L8BO:Y6 measured after fabrication (fresh) and aged for 384 h and 552 h, respectively.

Response letter | Figure 4. Normalized photovoltaic parameters of PTzBI-dF:L8BO measured after fabrication (fresh) and aged for 384 h and 552 h.

Response letter | Figure 5. Normalized photovoltaic parameters of PTzBI-dF:L8BO:Y6 measured after fabrication (fresh) and aged for 384 h and 552 h.

4. The statement of energy transfer between L8BO and Y6 is unclear to the readers. The spectral overlap between the crystalline/amorphous phase is fitted instead of directly measured. Therefore, the overlapping extent is questionable. On the other side, it is difficult to understand the RET from the amorphous cluster to the crystalline phase. During the measurement of photoluminescence, the excitation wavelength could excite both L8BO and Y6 and the PL spectra of L8BO:Y6 is the superposition of PL spectra of pristine L8BO and Y6. Therefore, it may not be proper evidence of RET. On page 10, the meanings of Φ_{RET} and Φ_{PL} are not clear.

Our response: We thank the referee for the helpful comment. There existed two narratives about “energy transfer process” in our manuscript. For the energy transfer between L8BO and Y6, the PL measurement could excite both L8BO and Y6 components in L8BO:Y6 blends, and the small disparity in the intensity might be aroused by the reduction of aggregation quench in Y6 because of its dilution by L8BO. To avoid misleading, we have removed the relevant statement and Figure in the revised manuscript.

The overlapping extent of measured absorption and PL spectra was shown in Response letter | Figure 6. For our current understanding, the spectra overlapping is a prerequisite for the RET from the amorphous cluster to the crystalline phase, as the red-shifted and structured ordered regions overlapped with blue-shifted and less structured amorphous bands [*ChemSusChem*, 2021, 14, 3590–3598]. As shown in Fig. 3c,

noticeable PL_A fitted from PL spectra showed increased intensity in both aged PTzBI-dF:L8BO and PTzBI-dF:Y6 blends, which was ascribed to the amorphous phase in blends. The used GIWAXS and supplemented TEM and RSoXS results also supported the occurrence of amorphous cluster in aged PTzBI-dF:L8BO and PTzBI-dF:Y6 blends. The ϕ_{PL} is the PL quantum yields in both the ordered and amorphous phases. Assuming similar ϕ_{PL} , the $\frac{\phi_{RET}}{\phi_{PL}}$ represents the relative energy transfer quantum yield from the amorphous to the ordered phase. We have specified this narrative in Page 11 in the revised manuscript.

Page 11: “Assuming similar PL quantum yields (ϕ_{PL}) in the ordered and amorphous phases, the relative energy transfer quantum yield from the amorphous to the ordered phase (ϕ_{RET} / ϕ_{PL}) can be used as a probe to detect the occurrence of larger clusters of the amorphous phase in films with the sketch shown in Supplementary Fig. 26.”

Response letter | Figure 6. Absorption and PL spectra for (a) L8BO, Y6 and L8BO:Y6 blends and (b) PTzBI-dF:L8BO, PTzBI-dF:Y6 and PTzBI-dF:L8BO:Y6 blends.

5. Capacitance-voltage/frequency measurements determine the trap density, and the density of states (DOS) vs. trap energy are displayed in figure 4g, 4h, and 4i. As shown in figure 4, compared with PTzBI-Df:L8BO and PTzBI-Df:Y6, the DOS center position of PTzBI-Df:L8BO:Y6 moves towards higher energy. What could be the possible reasons?

Our response: According to literature, a deeper trap energy might be related to the increasing effective interface band gap reported by previously work [*J. Phys. Chem. C*, 2014, 118, 21873-21883]. The DOS measurements reflect the density of deep states

because the capacitance measures the charge trapped in the active layer and the frequency determines what fraction of the trapped charge can respond to the ac field by thermal excitation to transport states. However, it cannot distinguish between electron and hole traps and measures the sum of both [*J. Phys. Chem. C*, 2016, 120, 22169-22178]. In our work, the DOS center position of PTzBI-dF:L8BO:Y6 moving to higher energy may be caused by the newly formed D/A interface after integrated L8BO:Y6 NFAs, and the formed interface can also conducted the long-term stability of OSCs. To clarify this point, we have added relevant sentences in Page 12 of the revised manuscript.

Page 12: “The DOS center position of PTzBI-dF:L8BO:Y6 moving towards higher energy may related to the changed interface proprieties in BHJ.”

Reviewer #3 (Remarks to the Author):

The manuscript “Mastering Microstructure Morphology of Non-fullerene Acceptors towards Long-term Stable Organic Solar Cells” by An et al. presented stable organic solar cells with high performances through a ternary blend strategy. Such phenomenon is observed in various reports (Nat. Mater. 2017, 16, 363; Adv. Mater. 2019, 1904601; ACS AMI, 2017, 9, 24, 20704; J. Mater. Chem. C. 2022, 10, 3207) with thermodynamic origins established (Adv. Energy Mater. 2018, 1702741). The authors employed diffraction, photonic spectroscopy, and thermal analysis to establish the nanoscale structure modification enabled by the mixing of two similar acceptor molecules. However, these analyses did not support the nanostructure illustrated by the authors due to either inappropriate application of the models or misinterpretation of the data. The main arguments of the manuscript including “better crystalline quality”, “attained thermodynamic equilibrium”, “suppress of amorphous aggregate to cluster” and “different thermodynamic diffusion behaviors” were not supported by the experimental results presented in the manuscript. Hence, I cannot support the publication of this manuscript in Nature Communications. Here are my comments in detail.

1. The authors named this manuscript “mastering microstructure ... organic solar cells”. In fact, the morphology of organic solar cells matters in nanoscale. The term “microstructure” refers to structures in micrometer scale. The author should revise the manuscript for this error.

Our response: We are grateful to the referee for the comments. As suggested by the referee, the morphology of organic solar cells matters in nanoscale. However, the idiomatic term about morphology is inconsistent in OPV community. The “microstructure” term was also widely used [*Nat. Commun.*, 2018, 9, 5335; *Joule*, 2021, 5, 2395-2407; *Energy Environ. Sci.*, 2019, 12, 1078-1087; *Chem*, 2021, 7, 2853-2854; *Adv. Funct. Mater.*, 2022, 32, 220178; *Adv. Energy Mater.*, 2022, 12, 2103940].

To avoid misleading, we revised the title of our manuscript name to “Mastering Morphology of Non-fullerene Acceptors towards Long-term Stable Organic Solar Cells” and modified relevant terms in the revised manuscript.

2. The authors discussed the gradual evolution of morphology of BHJ to the equilibrium state without considering the glass transition of the system, while there are numerous studies showed high glass transition can suppress such evolution. The authors should provide related discussions.

Our response: We greatly thank the referee for the helpful comment. The T_g is a critical parameter to predict the stability of OSCs. However, no detectable T_g can be observed in our DSC results (Supplementary Fig. 5). It is possible to increase the tendency for glass formation with high solidification rate to quench the material into a

glassy state [Adv. Energy Mater., 2018, 8, 1702741]. Here, we heated all DSC samples to the corresponding T_m and immediately cooled them in liquid nitrogen atmosphere (-77 K). As show in Response letter | Figure 7, still no detectable T_g transition can be observed. It is difficult to directly obtained T_g for some OSCs systems. To clarify this point, we have added relevant sentences in Page 6 of the revised manuscript.

Page 6: “The glass transition temperature (T_g) was also difficult to directly obtained from DSC profiles.”

Response letter | Figure 7. DSC first and second heating and cooling curves from (a) L8BO:Y6 (0.7:0.5) and (b) PTzBI-dF:L8BO:Y6 (1:0.7:0.5) blends. DSC first heating curves from (c) L8BO:Y6 blends and (d) PTzBI-dF:L8BO:Y6 blends with various ratios; all DSC samples were heating to the corresponding T_m and rapid cooling in a liquid nitrogen atmosphere.

3. Most processing procedures for OSC undergo a prolonged thermal annealing process at or above 100 °C, which is significantly higher than the service temperature of the OSCs. It is not suitable for stating “post treatments ... result in non-equilibrium morphology”. Furthermore, a true equilibrium morphology for a two-phase structure will be either a double layered system (i.e. oil on water, for immiscible) or solid solution (i.e. metal alloys, for miscible). It is against the fundamental requirements of OSC for pursuing equilibrium morphologies which cannot be BHJ. Can the authors clarify how the “attained thermodynamic equilibrium” is established in this work and how a molecularly mixed or two layered equilibrium structure providing such high

performances?

Our response: Thanks for the very helpful comment. As suggested by the referee, the narrative for “post treatments ... result in non-equilibrium morphology” and “attained thermodynamic equilibrium” in this manuscript is not suitable. To avoid misleading, we removed and modified corresponding description in the revised manuscript.

Page 3: The original narrative: “The manipulation of donor and acceptor pairing and pre/post treatments on the photoactive layer usually result in non-equilibrium morphology, which is not desirable to achieve long-term stability of OSCs.” was removed to avoid misleading.

Page 4: “Systematic investigation with grazing incidence wide-angle X-ray scattering (GIWAXS) and optical spectra revealed that tuning the NFA combinations of L8BO and Y6 not only gradually regulated film morphology, but also prevented the amorphous phase aggregation in BHJ thin film.”

4. The work is largely based on morphology study through spectroscopy. However, all the theoretical models of photonic spectroscopy established by numerous work of F. Spano, C. Silva, J. Clark and D. Neher are all based on P3HT which is a flexible molecule. The typical polymers used in current top performing OSCs are significantly more rigid (Nano Letters 2013, 13, 6, 2522) where models describing the photonic spectroscopy properties are yet established. The high throughput references the authors cited are based on matching device performances with spectrum of specific materials with theoretical models not being established. Hence, the authors cannot give such specific assignment to the “PL at NIR region” to “the clustered amorphous phase” without theoretical models established on specific materials used in this work or based on models established for P3HT which is a very different material. The discussions the authors offered on the spectroscopy measurements can only support “morphology can be fine-tuned by changing the blend ratio of NFAs”, but not “The amorphous phases in the PTzBI:L8BO:Y6 film could be effectively prohibited into cluster under thermal stress, which occurred in PTzBI:L8BO and PTzBI:Y6 films probed by optical spectroscopy” as in the discussion and introduction, and certainly not the images provided in Scheme 1 and S16.

Our response: We thank the referee for the comments. Optical spectrum is an effective auxiliary measurement to evaluate film morphology. Morphological information extracted from UV-vis-NIR and PL spectra in this manuscript are only semi-quantitative structural features, relying on Frank-Condon transitions and weak H aggregates (the so-called “Spano” model) [*ChemSusChem*, 2021, 14, 3590-3598]. Optical spectrum has many advantages, such as highly efficient, convenient, strong operability, and can be combined with high-throughput methods, which has been

utilized to different materials system. This method has been demonstrated to work well to NFAs, where morphology was found to be strongly correlated with electrical performance and degradation [Joule, 2021, 5, 495-506]. Moreover, recently work showed that morphological features can also be extracted from all-polymer systems [Nat. Energy, 2022, 7, 1180-1190]. It allowed us to unambiguously calculate the approximate structural features. Additional, we also evaluated our blends morphology with GIWAXS measurements identifying the ordered phases to support our conclusion. According to the reviewer’s suggestion, we conducted TEM and RSoXS measurements for aged PTzBI-dF:L8BO, PTzBI-dF:Y6 and PTzBI-dF:L8BO:Y6 blends to support the amorphous aggrate in our films (Supplementary Fig. 30). To clarify this issue, we have added relevant sentences in Page 11 and Supplementary Fig. 30 of the revised manuscript.

Page 11: “TEM and RSoXS characterizations were performed to evaluate the domain change in aged blends. Both PTzBI-dF:L8BO and PTzBI-dF:Y6 blends exhibited enlarged distribution of bright and dark regimes as well as obviously changed humps within the probed q range (Supplementary Fig. 30), which supported large amorphous cluster in films observed in optical spectrum, combined with the nearly identical GIWAXS patterns.”

Supplementary Fig. 30 | Morphology. (a) TEM images and (b) RSoXS averaged profiles for fresh and aged PTzBI-dF:L8BO, PTzBI-dF:Y6 and PTzBI-dF:L8BO:Y6 films.

5. The authors referred L8BO and Y6 with “different thermodynamic properties”. Do

the author mean thermodynamic properties such as heat capacitance and formation entropy? Can the authors provide specific values?

Our response: As suggested by the referee, the “thermodynamic properties” term used in this manuscript was not suitable. To avoid misleading, we carefully corrected the relevant terms with “thermal behavior”, “thermal proprieties” and “melting enthalpy” in the revised manuscript.

Abstract: “Here, we demonstrate stable OSCs ... feature similar structures yet different thermal and morphological proprieties”

Page 3: “Therefore, material-related thermal proprieties were introduced to estimate morphological optimization and long-term stability. By utilizing thermal factors, various NFAs can drive the morphology transitions from non-equilibrium towards equilibrium state via different approaches. If two or more similar NFAs with matched miscibility yet different thermal propriety and morphology are mixed together, a series of pseudo-new NFAs could be produced by varying the mixing ratios. ... Thus, it is of great importance to analyze their thermal behaviors and morphological changes as a function of the properties of NFAs, and to further explore the relationship between these factors and the long-term stability of efficient OSCs.”

Page 4: “It is advised to construct efficient and long-term stable OSCs via integrating two similar NFAs together to systematically regulate the thermal behavior and morphology. Here, we adopt the aforementioned strategy by employing two state-of-the-art NFAs (L8BO and Y6) featuring similar chemical structure (shown in Fig. 1a), yet different thermal proprieties and morphologies ...”

Page 6: “To explore the thermal behavior of L8BO:Y6 when combining with polymer PTzBI-dF, DSC measurements for neat PTzBI-dF film and PTzBI-dF:L8BO:Y6 blend films were performed (Supplementary Fig. 5d, e). Therefore, it is reasonable to analyze the thermal properties of PTzBI-dF and the two NFAs through the heat flow ΔH characters.”

Page 7: “Therefore, fine-tuning the thermal and morphological proprieties in PTzBI-dF:L8BO:Y6 was critical to modify charge transport as well as enhance J_{SC} and FF in devices. ... In this respect, controlling the thermal and morphological proprieties of NFAs in BHJ provides an effective strategy to alleviate energy loss and improve J_{SC} and FF values.”

Page12: “These combined results demonstrate that charge transport and collection were barely influenced on optimized PTzBI-dF:L8BO:Y6 film via fine-tuning thermal and morphology proprieties after long-term thermal stresses. ... The capacitance–voltage/frequency ($C - V/\omega$) measurements were used to examine the trap density difference aroused from the regulation of thermal and morphology proprieties in fresh and aged OSCs (Fig. 4d–i, Supplementary Fig. 35–36). ... Controlling the thermal and

morphological proprieties by two NFAs successfully demonstrated robust morphological stability and prohibited the accumulation of trap density, enabling much stable FFs and PCEs when combining the NFAs with the polymer PTzBI-dF.”

Discussion: “This work demonstrates the systematic regulation of the thermal behavior and morphology proprieties of NFAs, and reports an efficient strategy to simultaneously improve the PCE and thermal stability of NFA-based OSCs. The melting endotherm (ΔH) can be minimized through constructing two NFAs possessing different thermal proprieties with polymer donors. ... The combined regulation of thermal and morphological proprieties enables us to explore the relationship between these factors and the long-term stability of efficient OSCs. ... In summary, two NFAs (LB8O and Y6) with similar structure, yet different thermal proprieties and microscopic stacking morphologies were used to regulate the morphology of BHJ photoactive layer.”

6. Please provide the line cuts for the blend GIWAXS images and a demonstration of the fitting same as the spectroscopy analyses. The authors deduced “better crystalline quality” based on higher CCL values of pi-pi stacking for Y6 over L8BO. What do the authors mean by “better crystalline quality” do the author mean higher crystallinity or lower percentage of dislocations? Please provide values if available. Also the CCL values quoted in the main text is not consistent with Table S2.

Our response: The GIWAXS sector-averaged curves of blend film and a demonstration of detail multi-peak fitting are shown in Supplementary Fig. 3. The “better crystalline quality” represents the lower percentage of imperfection or dislocations that can disrupt the molecular packing. The CCLs of the lamellar stacking and π - π stacking reflections are summarized in Table S2. The CCL values quoted in the main text were also checked for consistence with Tables S2. To clarify this, we changed the following sentences in the manuscript.

Page 5: “The quality of crystallization can be evaluated by the crystalline coherence length (CCL), which is calculated through fitting the averaged I - q curves obtained from GIWAXS and using the Scherrer equation (Fig. 1b and Supplementary Fig. 2). A higher CCL represents a lower degree of imperfections and dislocations that can interfere the molecular packing in crystallites.”

Supplementary Fig. 3 | Molecular packing. The line cuts for the L8BO:Y6 blend and the demonstration of the curve fitting.

7. The authors compared the CCL of the pi-pi stacking L8BO and Y6. However, based on the single crystal structures of the two materials (Nature Energy 2021, 6, 605), the pi-pi stacking of Y6 containing two packing forms and L8BO containing three packing motifs. Hence, the resulting peak around 1.7 is the superposition of several peaks which is supported by the similar CCL calculated from the in-plane scatterings while such large CCL differences out-of-plane. Furthermore, such similarity of crystal sizes is supported by the RSoXS data where identical small angle scattering angle was found for all thin films. Hence, it is not valid to estimate CCL from the pi-pi stacking using FWHM obtained in scatterings containing more than one peaks in the case of this manuscript.

Our response: It is possible that in solution-processed thin films L8BO and Y6 may assume similar crystallization behaviors to those in single crystals. The CCL of the reflection peak at 1.7 \AA^{-1} was used to evaluate an average of CCLs for all π - π stacking. The RSoXS data reflects the correlation of domains with size scale beyond the CCLs of lamellar stacking. To avoid misleading, the original narrative of “crystal” was revised.

Abstract: “We find that the morphological features in terms of crystalline coherence length of fresh and aged films can be gradually regulated by the weight ratio of L8BO:Y6.”

Page 10: “These are combined signals from both polymer donor and NFAs. ... Each blend was estimated by CCLs of the lamellar stacking reflections (Supplementary Table 12).”

Page 11: “Combined with the greatly suppressed ΔH in PTzBI-dF:L8BO:Y6 blends, the mixed NFAs in PTzBI-dF:L8BO:Y6 blends showed the restrained aggregation, which was favorable for achieving stable morphology on both order and amorphous regions, and thus, improved device stability.”

Scheme 1: “In contrast, PTzBI-dF:L8BO:Y6 blends exhibited robust amorphous phase benefiting from the restrained aggregation movements and modified BHJ morphology via employing L8BO and Y6 NFAs.”

8. The thermal analysis presented in this work displayed a very interesting eutectic formation (Advanced Materials 2008, 20, 18, 3510). Hence, the thermal behavior of the blends cannot be simply compared with the enthalpy of fusion as the eutectic phase normally presents lower enthalpy of fusion for the presence of non-negligible surface energy. The lower enthalpy of fusion cannot be considered as the prove for “exhibit most robust morphological stability under thermal stress”. Furthermore, “L8BO and Y6 show different thermodynamic diffusion behaviors into PTzBI-dF” cannot be assessed by the thermal analysis. I suggest the authors to prepare bilayer structures and perform diffusion analysis using TOF-SIMS method to support this argument. (Macromolecules 2013, 46, 3, 1002) The phase behavior of similar acceptors blend is explored by Baran group with similar results, which I suggest the authors to go through. (ACS Energy Letter 2020, 5, 5, 1371; Advanced Functional Materials 2020, 2005462)

Our response: We greatly thank the referee for the helpful comment. According to the literatures recommended by the referee, we acknowledged that the decrease in enthalpy upon mixing NFAs may related to the form of a homogeneous amorphous mixture in similar acceptors blend [ACS Energy Lett., 2020, 5, 1371–1379; Adv. Funct. Mater., 2020, 30, 2005462]. In the revised manuscript, we removed relevant narratives for the relationship between “lower enthalpy of fusion” and “robust morphological stability”. The ΔH was only proposed a guiding criterion to select representative PTzBI-dF:L8BO:Y6 blends to evaluate the thermal stability of active layers, as the a well-mixed amorphous solid phase may occurred. We have specified or removed some narratives in Page 6 in the revised manuscript.

Page 6: “We focus on the blend ratio of 1:0.7:0.5 PTzBI-dF:L8BO:Y6 blends, as the mixed NFAs with similar structure may form a homogeneous amorphous mixture in blends, which also offers the best photovoltaic performance (cf. discussion on device performance below).” The original narrative: “The 1:0.7:0.5 blends may exhibit most robust morphological stability under thermal stress, as the minimized crystallization behaviors compared to the PTzBI-dF:L8BO and PTzBI-dF:Y6 blends.” was removed to avoid misleading.

According to the reviewer’s suggestion, we performed TOF-SIMS measurement on PTzBI-dF/L8BO and PTzBI-dF/Y6 bilayer structures (Supplementary Fig. 7). As shown in Supplementary Table 4, the $I_{(\text{CNO}^-)}/I_{(\text{CN}^-)}$ of PTzBI-dF exhibited two orders of magnitude higher than NFAs counterparts. Hence, the depth profile of PTzBI-dF was tracked with chemical group of CNO^-/CN^- as the label. The $\text{C}_{12}\text{H}_3\text{F}_2\text{N}_2\text{O}^-$ was used to identify NFAs. L8BO appeared in shorter sputter time after thermal-annealing (TA) compared to the as-cast film, while Y6 showed similar depth profiles before and after annealing (Supplementary Fig. 8). These findings suggested that L8BO and Y6 exhibit different thermal diffusion behaviors to PTzBI-dF. To clarify this issue, relevant

narratives has been modified or removed in Page 6, 7 and Methods part in the revised manuscript.

Page 6: L8BO and Y6 showed different thermal diffusion behaviors into PTzBI-dF, determined with time-of-flight secondary ion mass spectrometry (TOF-SIMS) (Supplementary Fig. 8).

Page 6: The original narrative: “Compared to the pristine L8BO and Y6 films, the ΔH values of PTzBI-dF:L8BO and PTzBI-dF:Y6 are suppressed by factors of 3 and 1.5, respectively, suggesting that L8BO and Y6 show different thermodynamic diffusion behaviors into PTzBI-dF.” was removed.

Page 7: “Collectively, integrating two NFAs with various ratios can effectively control the materials proprieties associated with thin-film morphology and molecular thermal behaviors.”

Time-of-Flight Secondary Ion Mass Spectrometry (ToF-SIMS)

Depth profiles of the bilayer samples were measured with a TOF-SIMS 5-100 instrument (ION-TOF GmbH, Germany). The instrument was equipped with a dual beam mode comprising of a Bi/Mn liquid metal ion gun (LMIG) and an argon gas cluster ion gun. A 5 keV Ar-cluster beam was used for depth profiling by sputtering through the film in 5 s intervals over an area of $300 \mu\text{m} \times 300 \mu\text{m}$. The central area within the sputtered region was analyzed using a 30 keV Bi^{3+} beam over an area of $100 \mu\text{m} \times 100 \mu\text{m}$. The obtained negative ion data was used for analysis.

The preparation of the bilayer samples involved several steps. First, solutions of PTzBI-dF (10 mg mL^{-1}), L8BO (14 mg mL^{-1}), and Y6 (14 mg mL^{-1}) in CF were separately prepared. NFAs was spin cast on pre-cleaned ZnO coated Si substrates at a spin-rate of 850 r min^{-1} . Neat PTzBI-dF film was spin cast on PEDOT:PSS coated glass at a spin-rate of 1000 r min^{-1} . Next, the PTzBI-dF film was then floated on DI water and transferred onto the NFAs/ZnO/wafer substrates, resulting in PTzBI-dF/NFA/ZnO/wafer bilayer samples. The samples were dried and can be post-thermally annealed at $100 \text{ }^\circ\text{C}$ for 5 minutes if necessary.

Supplementary Fig. 7 | TOF-SIMS. TOF-SIMS of PTzBI-dF/L8BO and PTzBI-dF/Y6 bilayer films under the as-cast and post thermal annealing (TA) conditions. The bilayer films were prepared on ZnO/Si wafer substrates.

Supplementary Fig. 8 | TOF-SIMS. Depth profiles for as-cast and TA bilayer structures of (a) ZnO/L8BO/PTzBI-dF and (b) ZnO/Y6/PTzBI-dF, respectively. TOF-SIMS results in Supplementary Table 4 reveals that both CNO^- and CN^- signals can be generated by PTzBI-dF and NFAs. In the case of PTzBI-dF, the intensity of CNO^- is higher than that of CN^- , whereas for NFAs, the opposite is observed, with the intensity of CNO^- being lower than that of CN^- . As a result, the CNO^-/CN^- intensity ratio is utilized to track the distribution of PTzBI-dF, while for the NFAs, the end group segment of $\text{C}_{12}\text{H}_3\text{F}_2\text{N}_2\text{O}^-$ is used as a label.

Supplementary Table 4 | The intensity of CNO^- and CN^- signals in TOF-SIMS of PTzBI-dF, L8BO, and Y6.

Neat films	Peak label	m/z	Intensity	$I_{(\text{CNO}^-)}/I_{(\text{CN}^-)}$
PTzBI-dF	CNO^-	42.0	8.43×10^5	2.26×10^{-1}
	CN^-	26.0	3.73×10^6	
L8BO	CNO^-	42.0	2.98×10^4	4.08×10^{-3}
	CN^-	26.0	7.30×10^6	
Y6	CNO^-	42.0	2.98×10^4	4.08×10^{-3}
	CN^-	26.0	7.30×10^6	

9. The authors should clarify how RSoXS data with almost identical scattering position can support “blending L8BO and Y6 can regulate the crystallinity and phase-separated length scale of BHJ thin films, opening an avenue to fine-tune nanostructure and photophysical properties of these blends.”

Our response: We thank the referee for the helpful comment. The regulation of crystallization was verified by the CCLs extracted from GIWAXS analysis, while tuning phase-separated length scales was supported by RSoXS measurements. To clarify this issue, we rephrased relevant sentences as follows.

Page 10, 11: “Hence, by blending L8BO and Y6, the CCLs in BHJ thin films can be regulated while maintaining phase-separated length scales, offering an effective method to finely adjust the morphology and photophysical properties of these blends.”

10. The author found a “significantly decreased crystallization behaviors of PTzBI-dF:L8BO:Y6 compared to the PTzBI-dF:L8BO and PTzBI-dF:Y6 blends.” The significantly decreased crystallization should be detrimental to the charge transport performance. However, they PTzBI-dF:L8BO:Y6 exhibited higher charge carrier mobilities than the binary blends. The authors should provide reasonable explanations on this.

Our response: In the revised manuscript, we removed relevant narrative for the relationship between “suppressed ΔH ” and “the crystalline ability”. The ΔH was only proposed a guiding criterion to select representative ratios for PTzBI-dF:L8BO:Y6 blend to evaluate the thermal stability of active layers, as the a well-mixed amorphous solid phase may occurred. The fitted CCLs of PTzBI-dF:L8BO:Y6 was gradually increased with the Y6 contents, extracted from GIWAXS analysis. Hence, both the μ_h and μ_e are increased compared to binary blends, suggesting that the hole and electron transport channels can be finely optimized by regulating the morphology of NFAs [*Energy Environ. Sci.*, 2020,13, 5039-5047; *Adv. Mater.*, 2018, 30, 1803045]. As the reviewer suggested, we have specified or de-emphasize some narrative in Page 6 and 7 in the revised manuscript.

Page 6: “The T_{m2} related to Y6 pinned at about 280 °C, suggesting the melting point could be easily affected even mixed with small amounts of L8BO (Supplementary Fig. 5c). ... but PTzBI-dF:L8BO:Y6 blends exhibited a broadened melting transition due to the NFAs. ... The ΔH_{mix} exhibits most suppress with 6.07 J g⁻¹ in 1:0.7:0.5 blend, indicating that the ΔH can be minimized through constructing two NFAs with different thermal proprieties (Supplementary Fig. 6). ... We focus on the blend ratio of 1:0.7:0.5 PTzBI-dF:L8BO:Y6 blends, as the mixed NFAs with similar structure may form a homogeneous amorphous mixture in blends, which also offers the best photovoltaic performance (cf. discussion on device performance below).” The original narrative: “The 1:0.7:0.5 blends may exhibit most robust morphological stability under thermal

stress, as the minimized crystallization behaviors compared to the PTzBI-dF:L8BO and PTzBI-dF:Y6 blends.” was removed to avoid misleading.

Page 7: “Enhanced and balanced charge mobilities were exhibited in PTzBI-dF:L8BO:Y6 OSCs, suggesting that the hole and electron transport channels can be finely optimized by regulating the morphology of NFAs.”

REVIEWER COMMENTS

Reviewer #1 (Remarks to the Author):

Most of my questions are well demonstrated in the revised manuscript. However, after reading the revision version, I think there are still two questions should be demonstrated more clearly before accepting:

1) The paper simplifies the morphology model by dividing them in crystalline region and inter-mixed region. Generally, the domain size characterized by other measurements is always many times larger than CCL, and the morphology is divided into donor-dominated phase, acceptor-dominated phase and inter-mixed phase. So, I think the author need to further demonstrate the aggregates' location according to our common phase division and then correlated it to the essential reason.

2) Second, combined the PL and Abs spectrum for the films based on single material (Y6 or L8BO) seems reasonable to illustrate the RET between amorphous and aggregates region. However, in their blend, there is RET between two acceptors, and the PL spectrum is not so reasonable. Further, for the ternary blend, ignoring the electron transfer or PL quenching to talk about RET, from my perspective, is needed to carefully consideration (Figure 3c). So, I think the analysis of PL spectrum is needed to further confirmation or explain more clearly.

Reviewer #2 (Remarks to the Author):

The authors answered all the comments raised, and the response are convincing, with literature and new results to support them. The suggestions made by all reviewers were addressed and revised in the manuscript. My main concern was with the optical spectroscopy analysis. Their TEM/RSoXS results on the fresh and aged samples could be linked to spectroscopy, and this approach is convenient for discussing morphology changes without advanced characterization techniques. Additionally, the authors provided device/stability results of another three polymer donors, demonstrating that the improved device stability is not from the donor materials but the Y6/L8BO combination. I am convinced by their findings.

Reviewer #3 (Remarks to the Author):

The authors have addressed the questions well. Therefore, this work can be accepted at this stage.

REVIEWER COMMENTS

Reviewer #1 (Remarks to the Author):

Most of my questions are well demonstrated in the revised manuscript. However, after reading the revision version, I think there are still two questions should be demonstrated more clearly before accepting:

1) The paper simplifies the morphology model by dividing them in crystalline region and inter-mixed region. Generally, the domain size characterized by other measurements is always many times larger than CCL, and the morphology is divided into donor-dominated phase, acceptor-dominated phase and inter-mixed phase. So, I think the author need to further demonstrate the aggregates' location according to our common phase division and then correlated it to the essential reason.

Our response: We greatly thank the referee for raising this important comment. Optical spectrum has many advantages, but it also presents some limitations in this work. According to literature [*ChemSusChem*, 2021, 14, 3590-3598], the order and amorphous phase of polymer donor can be unambiguously inferred from optical spectrum. Yet, the PTzBI-dF component was highly quenched with NFAs in our BHJ, and the donor-dominated phase cannot be well identified with the optical analysis. For the acceptor-dominated and intermixed phases, the optical factors in the NIR direction contained not only the pure NFAs, but also the intermixed phase for NFAs mixing with PTzBI-dF. Nevertheless, we mainly focused on the regulation of NFAs, and optical spectra analysis combined with other morphology characterization measurements could well support the illustration of crystalline region and inter-mixed region. However, it cannot distinguish the aggregates' location between donor/acceptor-dominated phase, and inter-mixed phase, and just represents the sum of aggregates' information from NFAs. To clarify this issue, we have added relevant sentences in Page 12 of the revised manuscript.

Page 12: "Generally, the morphology can be divided into donor/acceptor-dominated phases and inter-mixed phase. Here, we cannot distinguish the accurate aggregates' location according to the common phase division, considering the interpenetrated network of BHJ and highly quenched PL intensity of PTzBI-dF in blends. Nevertheless, the optical analysis can still represent the sum of aggregates' information from the amorphous phase."

2) Second, combined the PL and Abs spectrum for the films based on single material (Y6 or L8BO) seems reasonable to illustrate the RET between amorphous and aggregates region. However, in their blend, there is RET between two acceptors, and the PL spectrum is not so reasonable. Further, for the ternary blend, ignoring the electron transfer or PL quenching to talk about RET, from my perspective, is needed to

carefully consideration (Figure 3c). So, I think the analysis of PL spectrum is needed to further confirmation or explain more clearly.

Our response: We greatly thank the referee for the helpful comment, and agree with the referee that the optical analysis and discussion on the NFA blends should be further revised. This is also one limitation in the optical spectral analysis, which is due to the obvious spectral overlap between L8BO and Y6. Nevertheless, the film morphology of PTzBI-dF:L8BO:Y6 could be also well confirmed with TEM and RSoXS results. To illustrate this issue, we have de-emphasized the optical probes to PTzBI-dF:L8BO:Y6 blends and modified relevant sentences in Page 12 in the revised manuscript.

Page 12: “Considering the obvious overlap of L8BO and Y6 optical spectrum, potentially RET between two NFAs may disturb the information obtained from the PTzBI-dF:L8BO:Y6 film. ... The PTzBI-dF:L8BO:Y6 blend exhibited stable film morphology.”

Page 12: The sentence was removed to avoid misleading.: “The PTzBI-dF:L8BO:Y6 film exhibited only slightly decay of ϕ_{RET} / ϕ_{PL} .”

Reviewer #2 (Remarks to the Author):

The authors answered all the comments raised, and the response are convincing, with literature and new results to support them. The suggestions made by all reviewers were addressed and revised in the manuscript. My main concern was with the optical spectroscopy analysis. Their TEM/RSoXS results on the fresh and aged samples could be linked to spectroscopy, and this approach is convenient for discussing morphology changes without advanced characterization techniques. Additionally, the authors provided device/stability results of another three polymer donors, demonstrating that the improved device stability is not from the donor materials but the Y6/L8BO combination. I am convinced by their findings.

Our response: We greatly thank the reviewer for the positive evaluation of our revised manuscript, and are very delighted to see that the reviewer is satisfied with our revision. We had carefully carried out additional characterization measurements to link and support the spectral analysis. In addition, the stability results of other polymer donors also exhibited obvious improvement in thermal stability. We are grateful to reviewer for his/her effort reviewing our manuscript and his/her positive feedback.

Reviewer #3 (Remarks to the Author):

The authors have addressed the questions well. Therefore, this work can be accepted at this stage.

Our response: We are grateful to the reviewer for his/her very positive evaluation and the effort spent reviewing our manuscript.

REVIEWERS' COMMENTS

Reviewer #1 (Remarks to the Author):

I think the expressions after revision are more cautious than before, and I have no further questions for this paper. I recommend its acceptance.